# PREDICTING LLM REASONING PERFORMANCE WITH SMALL PROXY MODEL

**Woosung Koh**$^{\nabla\spadesuit}$**, Juyoung Suk**$^{\nabla\spadesuit}$**, Sungjun Han**$^{\nabla\star}$**, Se-Young Yun**$^{\spadesuit\star}$**, Jamin Shin**$^{\nabla\star}$
$^{\nabla}$Trillion Labs, $^{\spadesuit}$KAIST AI
$^{\star}$Correspondence to: yunseyoung@kaist.ac.kr, {sungjun.han, jay}@trillionlabs.co

## ABSTRACT

Given the prohibitive cost of pre-training large language models, it is essential to leverage smaller proxy models to optimize datasets before scaling up. However, this approach becomes challenging for reasoning capabilities, which exhibit *emergent* behavior that only appear reliably at larger model sizes, often exceeding 7B parameters. To address this, we introduce RBRIDGE, showing that small proxies ($\leq$1B) can effectively predict large-model reasoning by aligning more closely with **(1)** the pre-training objective and **(2)** the target task. RBRIDGE achieves this by weighting negative log-likelihood with task alignment, using reasoning traces from frontier models as gold labels. In our experiments, RBRIDGE **(i)** reduces dataset ranking costs by over $100\times$ relative to the best baseline, **(ii)** achieves the strongest correlation across six reasoning benchmarks at 1B to 32B scale, and **(iii)** zero-shot transfers predictive relationships across pre-training datasets at 1B to 7B scale. These findings indicate that RBRIDGE offers a practical path for exploring reasoning-oriented pre-training at lower cost.

🤗 **RBRIDGE**

## 1 INTRODUCTION

Pre-training modern language models at scale requires enormous computational and data resources, making it infeasible to exhaustively explore pre-training design choices directly at large scale (Radford et al., 2018; 2019; Brown et al., 2020; Dong et al., 2019; Hoffmann et al., 2022a; Cottier et al., 2024; Hu et al., 2024; Khandelwal et al., 2024; Han et al., 2025). In response, leveraging smaller models as a proxy for larger model performance has been a key direction through the establishment of empirical scaling laws for prediction (Kaplan et al., 2020; Hoffmann et al., 2022b) or derivation of pre-training dataset rank invariance across scale (Magnusson et al., 2025).

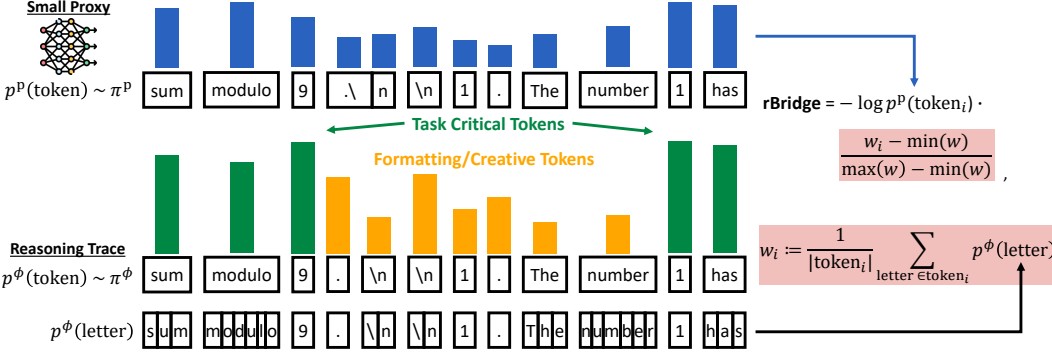

Figure 1: Schematic overview of RBRIDGE, which is used to predict and rank performance at much larger model size. We use a frontier model $\pi^{\phi}$'s reasoning trace (Wei et al., 2022b) as the gold label $Y^*$ and compute weighted NLL for evaluation. Each token $i$'s NLL is weighted by the frontier model's confidence in that token (MinMax normalized). To handle tokenizer mismatches between proxy and frontier models, we compute weights at the letter level and average within tokens.

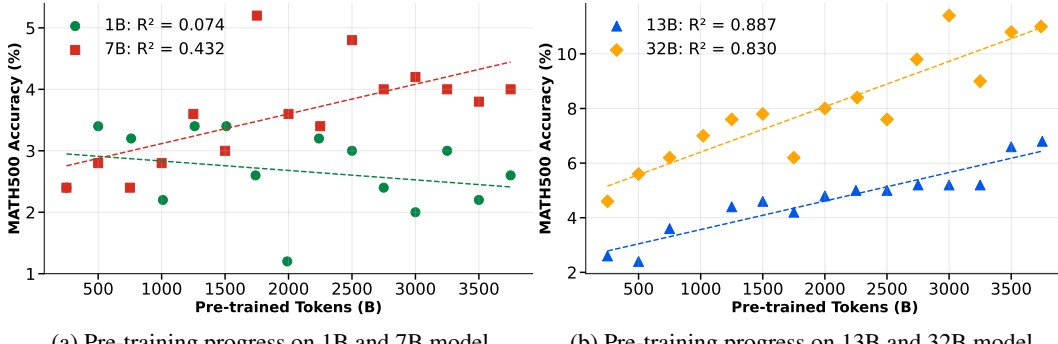

(a) Pre-training progress on 1B and 7B model     (b) Pre-training progress on 13B and 32B model

Figure 2: Using MATH500 as an example benchmark, given the same data source OLMo-Mix-1124 (OLMo et al., 2024), smaller models exhibit more noise and get the direction wrong, making it challenging to use smaller models to proxy larger model performance. $R^2$ values are derived from linear curve fitting. Extended visualization across other reasoning benchmarks are available in Appendix A.

However, the literature on the *emergence* of reasoning performance as we scale model size (Wei et al., 2022a; Almazrouei et al., 2023; Du et al., 2024) suggests that there may be a limit to how small theses proxy models can be. They demonstrate that reasoning capabilities only appear when models are sufficiently large in size. Du et al. (2024)'s granular study demonstrates *random* accuracy on small scale models of size 300M - 3B benchmarked on reasoning tasks like MMLU (Hendrycks et al., 2021a) and GSM8K (Cobbe et al., 2021). Contrarily, other non-reasoning benchmarks like TriviaQA (Joshi et al., 2017) and HellaSwag (Zellers et al., 2019) show smooth signs of improvement even at small scale.

We further visualize this challenge of using small models to proxy large model especially for reasoning in Fig. 2. While larger models exhibit stable Accuracy (Acc.) improvement (Fig. 2b), smaller models are highly noisy, and in the case of the smallest 1B model, sloping in the wrong direction (Fig. 2a).

Due to this limitation, practitioners are often constrained to relatively larger proxy models up to 15B to capture reasoning performance, which incurs substantial computational and economic costs (Grattafiori et al., 2024; DeepSeek-AI et al., 2025). For example, a single training run of a 7B model with 500B tokens can reportedly exceed 50K USD in cost (Han et al., 2025).

**Contribution.** To bridge the evaluation scheme at small proxy to large target scale, we first analyze limitations of past approaches (§ 3.1). Our analysis uncovers that existing methods fail to (§ 3.1 (1)) align with the pre-training objective, and (§ 3.1 (2)) align with the target task. **(1)** Alignment with the pre-training objective is required as small pre-trained models *lack strong generalization capabilities*. **(2)** Ensuring that the evaluation scheme is aligned with the target task is necessary to fulfil our *ultimate* goal of proxying task performance at large scale. To achieve **(1, 2)**, we use frontier-model generated gold reasoning traces (Wei et al., 2022b) for negative log-likelihood (NLL) (Fig. 1). Then, we further task alignment at the token-level by automatically weighting tokens based on their level of task-alignment (Fig. 1). We empirically validate our method RBRIDGE (§ 4) in the following five ways:

1. On a pre-training dataset ranking benchmark with target 1.2B scale, RBRIDGE achieves **80.8%** decision accuracy across **25** pre-training datasets, outperforming **5** baselines, reducing dataset ranking compute cost by at least **100.2×** against the best .

2. RBRIDGE achieves best average proxy (1B) to target (13B, 32B) relationship across **6** benchmarks (mathematics, science, engineering, commonsense, and coding tasks), against **6** baselines.

3. RBRIDGE achieves best average 1B → 13B relationship after a supervised fine-tuning (SFT) stage at target scale across **4** benchmarks, against **6** baselines.

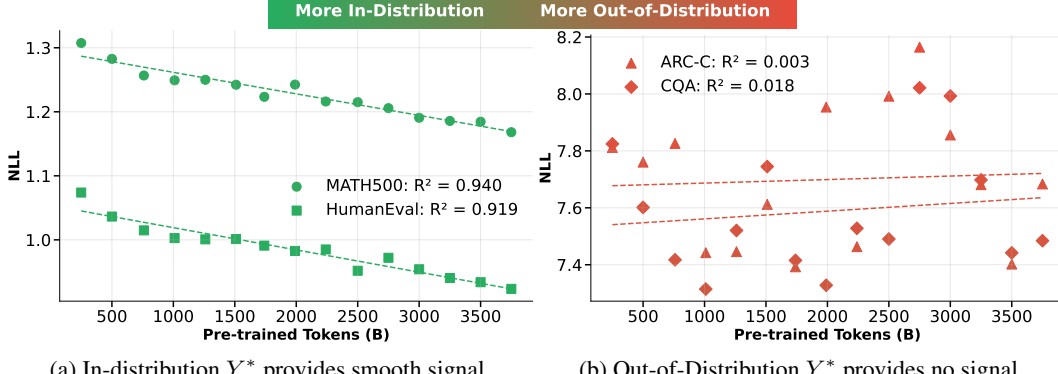

(a) In-distribution $Y^*$ provides smooth signal.  (b) Out-of-Distribution $Y^*$ provides no signal.

Figure 3: When evaluating a 1B pre-trained model with next token prediction, $\pi^{\mathrm{p}}(y_\tau | x, y^*_{<\tau})$, whether the target $Y^*$ is in-distribution becomes important. All visualized benchmarks demonstrate smooth improvements at larger ($13\times$, $32\times$) scale with target metric Acc./p@k. For clarity, we visualize the benchmarks in our empirical study with the two smallest and largest average NLL values.

4. RBRIDGE outperforms proxy models **7 - 13**$\times$ larger using the target metric (e.g. Acc., Pass@K).

5. Finally, we demonstrate that RBRIDGE-target relationship on one pre-trained dataset can be **zero-shot transferred** to an alternative dataset for low-error performance prediction and ranking using a **fraction** of experimental compute cost.

## 2 PROBLEM SETTING

Let $\pi^{\mathrm{p}}, \pi^{\mathrm{t}}$ denote small proxy and large target model, respectively. The target metric $\mathrm{metric}^{\mathrm{t}}$ (e.g., Acc., Pass@k) is fixed. Our objective is to design a proxy evaluation $\mathrm{metric}^{\mathrm{p}}$ such that considering $f : \mathrm{metric}^{\mathrm{p}} \mapsto \mathrm{metric}^{\mathrm{t}}$, find $\mathrm{metric}^{\mathrm{p}} := \max_{\mathrm{metric}} \mathrm{corr}(\mathrm{metric}(\pi^{\mathrm{p}}), \mathrm{metric}^{\mathrm{t}}(\pi^{\mathrm{t}}))$. We discuss the $\mathrm{corr}(\cdot)$ function we use in future sections. That is, improvements observed at proxy scale should reliably predict improvements at target scale. We denote scaling experiments as $n \to m$, where $n, m$ is the proxy, target model size, respectively; e.g., $1B \to 32B$. Performance changes can arise from either **(I)** varying the training dataset at fixed data size, or **(II)** varying the training data size given a fixed dataset. This enables two key applications: **(I)** comparing alternative training datasets without training large models on each, and **(II)** predicting whether scaling up training data (e.g., $3000B \to 3500B$ tokens) is worthwhile. This is important because it enables us to predict the return on investment for large-scale training before committing resources. A practical proxy evaluation scheme must therefore be reliable at *small scale* and achieve *high correlation*.

## 3 BRIDGING SMALL AND LARGE MODEL SCALE EVALUATION

We find that small models become strong proxies for large models when achieving alignment along two axes: alignment with the pre-training evaluation objective (**§ 3.1 (1)**), and alignment with the target task (**§ 3.1 (2)**). In response, we introduce our method, RBRIDGE (**§ 3.2**).

### 3.1 PRIOR APPROACH LIMITATION

**(1) Evaluation Objective Misalignment.** We find that the first limitation of existing approaches is their lack of evaluation objective alignment with $\pi^{\mathrm{p}}$. This is required as small pre-trained models *lack strong generalization capabilities*. Concretely, **(a)** Acc. and Pass@K (p@k) is misaligned with the pre-training objective function, and **(b)** even if we use an objective function aligned evaluation scheme like NLL, we must stay aligned with the pre-trained model's distribution.

**(1.a) Misaligned Evaluation Metric.** Existing target metrics like Acc./p@k are misaligned with the proxy model's next token prediction (NTP) NLL

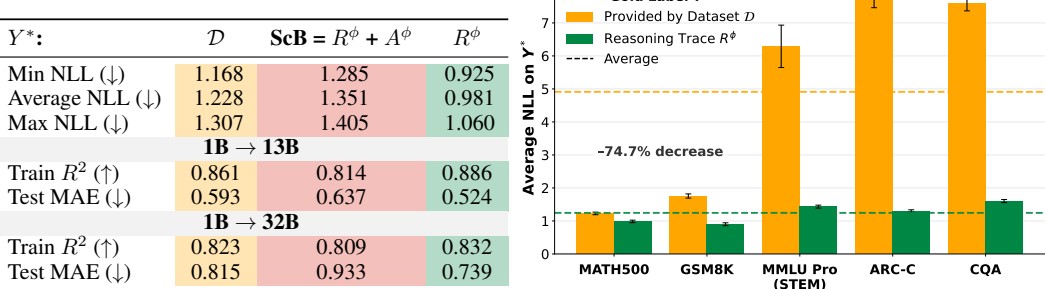

| $Y^*$: | $\mathcal{D}$ | $\textbf{ScB} = R^\phi + A^\phi$ | $R^\phi$ |
|---|---|---|---|
| Min NLL ($\downarrow$) | 1.168 | 1.285 | 0.925 |
| Average NLL ($\downarrow$) | 1.228 | 1.351 | 0.981 |
| Max NLL ($\downarrow$) | 1.307 | 1.405 | 1.060 |
| **1B → 13B** | | | |
| Train $R^2$ ($\uparrow$) | 0.861 | 0.814 | 0.886 |
| Test MAE ($\downarrow$) | 0.593 | 0.637 | 0.524 |
| **1B → 32B** | | | |
| Train $R^2$ ($\uparrow$) | 0.823 | 0.809 | 0.832 |
| Test MAE ($\downarrow$) | 0.815 | 0.933 | 0.739 |

Table 1: We observe a direct relationship between degree of in-distribution (as presented as NLL) and performance. Best to worst in each row labeled as green, orange, red.

Figure 4: Using reasoning trace $R^\phi$ over benchmark test set's $Y^*$ significantly reduces NLL, suggesting that $R^\phi$ is more in-distribution. Error bars indicate one standard deviation.

Consider Fig. 3a, where NLL shows smooth improvement with correct slope at 1B on MATH500 while Acc. in Fig. 2 is noisy and sloping the wrong direction.

**(1.b) Not All NLL are Equal: Distributional Alignment.** Furthermore, we find that the quality of the NLL signal hinges on the gold label $Y^*$. We define gold label as the string used for NLL on our proxy model. We observe that it should be closer to the pre-training distribution $p(y_\tau|x, y^*_{<\tau}) \sim \pi^p$ where $x$ is the input, and $\tau$ is the decoded token. Following Arora et al. (2021), Gonen et al. (2023), we measure how in-distribution (ID) $Y^*$ is via NLL, $-\log(p(Y^*))$. The more out-of-distribution (OOD), the more lacking in signal the proxy model becomes (Fig. 3). Benchmarks with an ID $Y^*$ (lower NLL) show smooth and predictable progress at smaller model scale (1B) (Fig. 3a). Contrarily, benchmarks with OOD $Y^*$ (higher NLL) is at least as noisy as the target metric Acc. (Fig. 3b).

We also observe this distributional misalignment in ScalingBench (ScB; Xiao et al. (2024); Team (2025)). ScB proposes we set the reasoning trace $R^\phi$ (Wei et al., 2022b) and the final answer label $A^\phi$ of a frontier model $\pi^\phi$ as $Y^*$. ScB's gold labels include formatting artifacts like "\n", "Final Answer:", and "I hope it is correct." that rarely appear in pre-training data, making them OOD, hurting proxy performance (Tab. 1). The proxy evaluation protocol is detailed in **§ 4.1 (ii)**.

**(2) Target Task Alignment of Gold Label $Y^*$ and at the Token Level.** Secondly, we hypothesize that the evaluation scheme should be task aligned. Here, target task refers to, e.g., correctly solving math problems for GSM8K (Cobbe et al., 2021), and MATH500 (Hendrycks et al., 2021b). Ensuring that $Y^*$ is aligned with the target task is necessary to fulfilling our *ultimate* goal of proxying task performance at large scale. While we could achieve maximal ID by setting the greedy decoded tokens of $\pi^p(y_\tau|x, y^*_{<\tau})$ as $Y^*$, this would provide no signal, as it would fail to be task aligned.

We further observe that standard NLL on $Y^*$ does not distinguish between tokens that are important for task alignment, and others that are less task critical. Consider, Fig. 1, where frontier model $\pi^\phi$ decoding produces both **task critical** and **formatting/creative** tokens. For example, newlines and numbering are not essential, while steps like "sum modulo 9" are crucial. To further achieve task alignment, tokens should not be arbitrarily given equal weight as is in NLL.

### 3.2 rBridge: Improving Evaluation Objective Alignment and Task Alignment

**Reasoning Trace $R^\phi$ as $Y^*$.** We show that using only the reasoning trace $R^\phi$ of a frontier model $\pi^\phi$ as $Y^*$ satisfies both being **(1)** ID and **(2)** task alignment. **(1)** We hypothesize that $R^\phi$ is more distributionally aligned (ID) with the pre-training distribution comprised predominantly of a collection of continuous long texts (Penedo et al., 2024; Kydlíček et al., 2025). We verify this empirically in Fig. 4, where we visualize average NLL across 250 to 3750B trained tokens on a 1B model, on the provided $Y^*$ by the benchmark dataset $\mathcal{D}$ against the frontier model $\pi^\phi$ generated $R^\phi$. We observe an average NLL decline of 74.7% when using $R^\phi$ across five reasoning benchmarks, providing supportive evidence that $R^\phi$ is more ID. **(2)** $R^\phi$ is well aligned with the target task as $R^\phi$

is the reasoning trace that leads to the correct final answer for Acc./p@k. Intuitively, evaluating how well the model $\pi^p$ reasons towards the final answer is a good ID proxy for achieving the target Acc./p@k. Empirically, using only $R^\phi$ achieves the best relationship on MATH500 which we are able to replicate as the exact ScB $Y^*$ gold labels have been released for MATH500 (Tab. 1).

**Weighted NLL on $R^\phi$'s Tokens for Further Task Alignment.**  RBRIDGE takes a final step for further task alignment by weighting each token by its level of task-alignment. We hypothesize that frontier model token probability $p^\phi(\text{token}_i) \sim \pi^\phi$ provide automatic task-alignment weights:

$$\text{RBRIDGE NLL}(\text{token}_i) := \underbrace{-\log(p^p(\text{token}_i))}_{\text{standard NLL}} \cdot \underbrace{\frac{1}{|\text{token}_i|} \sum_{\text{letter} \in \text{token}_i} p^\phi(\text{letter})}_{\text{Automatic tokenizer-agnostic task-alignment weight}} . \quad (1)$$

Our method weighs each token $i$'s NLL by the frontier model's confidence in that token $p^\phi(\text{token}_i)$. To handle different tokenizers, we average letter-level probabilities within each token. Finally, we apply MinMax normalization (Witten & Frank, 2002) on the weight factor in Eq. (1) to amplify the effect. Consider Fig. 1 for an intuitive visualization of Eq. (1) with the MinMax normalized form unpacked in full. The pseudocode and prompt is available at Appendix B.

## 4 EMPIRICAL STUDY

Our experiments are organized in three stages: **(i)** first, we show that RBRIDGE can be used to rank datasets from proxy scale of $<$100M to target scale 1.2B, **(ii)** next, we examine the RBRIDGE-target relationship across different amounts of training data at 1B to 32B scale, and **(iii)** finally, demonstrate that the relationship derived in **(ii)** can be effectively transferred to a different dataset, allowing us to predict *and* rank large-scale datasets' reasoning performance at a fraction of the cost.

### 4.1 EXPERIMENTAL PROTOCOL

All experimental settings are set *a priori*, and all methods are evaluated in the same manner. Detailed experimental details are available in Appendix C.

**(i) Dataset Ranking at $<$100M $\rightarrow$ 1.2B.**  We evaluate whether RBRIDGE scores from proxy models can effectively rank pre-training datasets for target model performance. Following DataDecide's protocol (Magnusson et al., 2025), we rank 25 datasets using proxy models and assess alignment with target model rankings. We measure effectiveness using Decision Accuracy (DAcc.; fraction of correctly ordered dataset pairs; Magnusson et al. (2025)) and Kendall's Tau correlation. Compute cost is measured as FLOPs $= 6ND$, where $N$ is model size, and $D$ is dataset size, following Kaplan et al. (2020).

**Setup.** Proxy models range $n \in [3.7\text{M}, 97.9\text{M}]$, with a 1.2B target model. As discussed in § 1, noisy benchmarks such as MATH500 and MMLU Pro are excluded at this scale. Instead, we use ARC-C (Clark et al., 2018) and CQA (Talmor et al., 2019), which yield stable cloze-form (CF) accuracy (Gu et al., 2025) when averaged across three pre-training seeds (Magnusson et al., 2025). Reported results are the average over these two benchmarks.

**Baselines.** Alongside RBRIDGE, we compare five *dataset-ranking* metrics commonly used with proxy models: Correct Probability, Normalized Correct Probability, Total Probability, Margin, and CF Accuracy (Xie et al., 2023; Liu et al., 2025; Magnusson et al., 2025).

**(ii) Proxy-target Relationship Across Different Amounts of Training Data at 1B $\rightarrow$ 32B.**  We test whether proxy-scale metrics can predict changes in target-scale Acc. and p@k across different training data sizes. Each data point corresponds to a specific data size from the same source (Fig. 5). Following Che et al. (2018), Lee et al. (2020), we evaluate using 5-fold cross validation on the setup visualized in Fig. 5, reporting average train $R^2$ and test Mean Absolute Error (MAE; Chai & Draxler (2014)). Curve fitting selects the best function based on train $R^2$ from our hypothesis space: linear, quadratic, exponential, and logarithmic. This hypothesis space was defined *a priori* to avoid overfitting. FLOPs

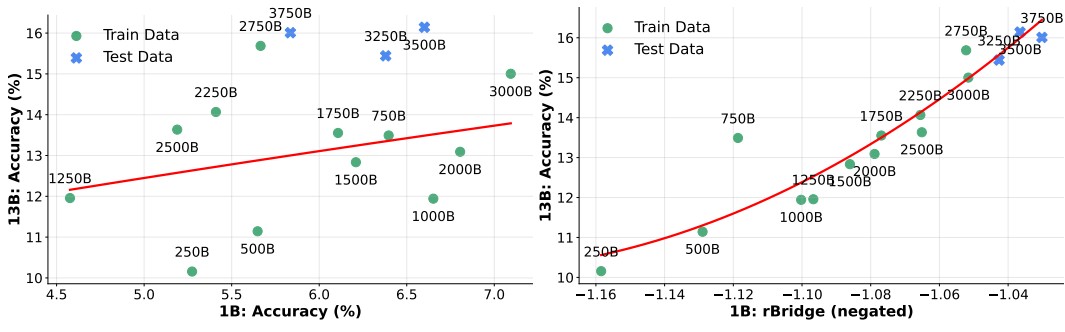

(a) Target metric Acc. as the proxy metric at 1B.

(b) RBRIDGE as the proxy metric at 1B.

Figure 5: Example visualization from a fold of proxy-target relationship study at 1B → 13B on MMLU Pro (STEM). Each data point represents equal trained tokens for the proxy and target model. Extended visualizations for all other benchmarks available in Appendix D.

**Setup.** We examine model scales of 1B → 13B and 1B → 32B, pre-trained on 250B to 3750B tokens in 250B token intervals using the OLMo-Mix-1124 dataset (OLMo et al., 2024). Evaluation spans six reasoning benchmarks: GSM8K, MATH500, ARC-C, MMLU Pro (STEM subset; Wang et al. (2024)), CQA, and HumanEval (Chen et al., 2021).

**Baselines.** We cover six alternative metrics that examine the relationship of small and large models given the *same* data source, and metrics that have shown non-emergent continuity at small scale. First, we include the most naïve approach, the target metric: Accuracy and Pass@1 (Acc./p@1; Kulal et al. (2019)). Second, we include intermediate supervised fine-tuning (iSFT; Snell et al. (2024)) which demonstrates that including a SFT stage throughout the intermediate pre-training checkpoints helps target metric (Acc./p@1) be used as signals. Third is Token Edit Distance (TED; Schaeffer et al. (2023)) which argue that emergence occurs due to discontinuous metrics, and correspondingly proposes their continuous metric. Fourth is Model Probability of Correct Answer (MPCA) as demonstrated in Schaeffer et al. (2023), Snell et al. (2024) as a continuous metric. Last is ScB which visualizes the relationship between ScB and target metric using *numerous* smaller proxy models (similar to the scaling law literature). As mentioned in § 3, we use our proposed improved version of ScB, $R^\phi$, as there were insufficient details to replicate ScB on all of our benchmarks.

**(iii) Zero-shot Functional Relationship Transfer Across Dataset at 1B → 7B.** Finally, we ask: *if we fit an empirical function on a single pre-training dataset $\mathcal{D}_{pre}$ as in* **(ii)**, *can this function transfer directly to a different dataset $\mathcal{D}'_{pre}$?* Concretely, once we fit Acc./p@k = $f($RBRIDGE$)$ on $\mathcal{D}_{pre}$, can this learned function transfer zero-shot (i.e., with no additional fitting) to $\mathcal{D}'_{pre}$ where $\mathcal{D}_{pre} \neq \mathcal{D}'_{pre}$? If so, we could predict the performance (Acc./p@k) of $\mathcal{D}'_{pre}$ at target scale using only the proxy model $\pi^p$, reducing compute by a factor of $m/n$, recalling that $m$, $n$ are the target and proxy sizes, respectively. This allows us to both estimate performance for any number of additional datasets and rank them at a given pre-training size $D'_{pre} \in \mathbb{N}^+$, simply by inputting the RBRIDGE score in $f(\cdot)$ after training on $D'_{pre}$ tokens from $\mathcal{D}'_{pre}$.

**Setup.** We curate an additional pre-training dataset $\mathcal{D}'_{pre}$ described in Appendix C. Then, we examine how accurate such function as described in **(ii)**, zero-shot transfers from the OLMo-Mix-1124 to our alternative dataset at 1B → 7B at 1T tokens. While larger-scale studies across more model sizes and pre-training datasets would be ideal, training an additional 1B and 7B model for 1T tokens required *thousands* of H100 hours, making further comparisons prohibitively expensive.

## 4.2 RESULTS

**(i) Over 100× Dataset Ranking Compute Saving at <100M → 1.2B.** RBRIDGE significantly improves DAcc. given equivalent compute (Fig. 6a). For example, at the most compute-efficient point (3.7M model trained on 87.3M tokens), RBRIDGE achieves up to 27% higher DAcc. than baseline metrics. This is remarkable as at this proxy compute level, CF Accuracy and Norm Correct

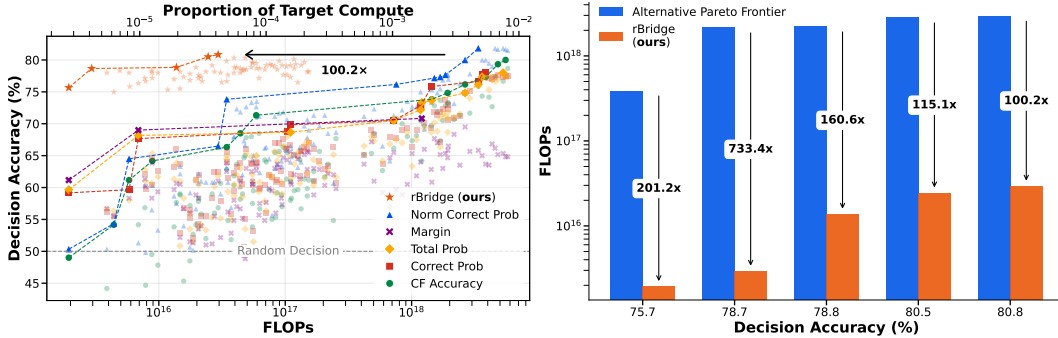

(a) Decision Accuracy results across 3.7 - 97.9M proxy model size, and across intermediary checkpoints. (b) RBRIDGE saves FLOPs by a factor of 100.2× to 733.4×.

Figure 6: RBRIDGE improves the pareto frontier in pre-training dataset ranking for 1.2B target model size. Values are averages aggregated across ARC-C and CQA. For intuitive reference, the two most compute efficient points in RBRIDGE's pareto frontier is (1) 3.7M model size trained on 87.3M tokens, and (2) 6M model size trained on 81.6M tokens.

Prob displays random decision levels ($\sim$ 50%). To achieve the same level of DAcc., RBRIDGE uses 100.2× to 733.4× less FLOPs (Fig. 6b). Kendall's Tau results are reported in Appendix D. As the two evaluation metric are intimately related, the results are highly similar.

**(ii) Consistently Strongest Relationship at 1B→13B, 32B.** In 1B →13B, RBRIDGE achieves the best $R^2$ and MAE in 10/12 cases, and ranks near the top in the remaining two (Tab. 2). Similar performance occurs in 1B→13B+SFT where we include a single epoch SFT stage at target scale, and 1B→32B. For 1B→13B+SFT and 1B→32B, granular benchmark-level results appear in Appendix D. Aligned with the literature on emergence, discontinuous metrics (Acc./p@1, iSFT) demonstrated worst performance. The remaining continuous metrics performed better, with RBRIDGE's state-of-the-art performance achieving train $R^2$ of 0.826 - 0.874 and test MAE of 1.304 - 1.481.

Additionally, even as we increase the proxy model size by 7×, 13×, the target metric fails to out-perform RBRIDGE (Fig. 7a). Notably, RBRIDGE's and best alternative box-and-whisker's box does not overlap, suggesting a meaningful improvement. To understand where these gains originate, we ablate the components of RBRIDGE in Fig. 7b, showing that each part of the RBRIDGE NLL (Eq. 1) plays an important role to its effectiveness across all three experimental settings. Note that we have an ablation study on going from NLL to ScB to $R^\phi$ in § 3.2, therefore, skip this part here.

**(iii) Successful Zero-shot Functional Relationship Transfer at 1B→7B.** We demonstrate that functions fitted on dataset $\mathcal{D}_{\text{pre}}$ with RBRIDGE (from **(ii)**) can zero-shot transfer to alternative pre-training dataset $\mathcal{D}'_{\text{pre}}$, reducing additional experimental compute cost by factor $\frac{m}{n}$ (Tab. 3). In our experiment setting of 1B→7B, this yields a 7× compute reduction since only a proxy model of $\frac{1}{7}$ the target size is required. The transferred function achieves low MAE of 0.043 - 1.417 across most benchmarks (one outlier at 9.716), outperforming $R^\phi$. For dataset ranking using predicted accuracy, RBRIDGE achieves perfect 5/5 performance versus $R^\phi$'s 4/5. This demonstrates that improved function fitting from **(ii)** translates to superior zero-shot transfer performance across datasets.

## 5 DISCUSSION

**Additional Related Work.** Zhang et al. (2024) uses proxy benchmarks with small proxy models to predict pre-training performance of larger target models. However, their approach suffers from several limitations. First, it is computationally expensive, requiring global search over 42 benchmarks and 34 models. Second, the required size of the search space (number of benchmarks and models) lacks principled justification. Third, using fundamentally different test distributions introduces suboptimal noise, as we empirically demonstrate in Appendix E.

Xie et al. (2023) and Liu et al. (2025) leverage small proxy models (1 to 280M parameters) to optimize pre-training data *mixture weights* for larger target models ($\sim$7B), operating under the as-

| Benchmark | Method: | Acc./p@1 | iSFT | TED | MPCA | NLL | $R^\phi$ | RBRIDGE |
|---|---|---|---|---|---|---|---|---|
| **1B → 13B (Pre-to-Pre)** | | | | | | | | |
| GSM8K | Train $R^2$ | 0.402 | 0.385 | 0.558 | 0.116 | 0.853 | **0.947** | 0.944 |
| | Test MAE | 5.189 | 4.848 | 5.961 | 84.006 | 3.210 | 1.837 | **1.751** |
| MATH500 | Train $R^2$ | 0.127 | 0.076 | 0.213 | 0.025 | 0.861 | 0.864 | **0.890** |
| | Test MAE | 1.276 | 1.008 | 1.134 | 3.943 | 0.593 | 0.526 | **0.525** |
| ARC-C | Train $R^2$ | 0.200 | 0.166 | 0.547 | 0.319 | 0.166 | 0.950 | **0.969** |
| | Test MAE | 7.287 | 8.750 | 6.197 | 25.970 | 7.058 | 1.546 | **1.246** |
| MMLU Pro | Train $R^2$ | 0.167 | — | 0.199 | 0.200 | 0.207 | **0.897** | **0.897** |
| (STEM) | Test MAE | 1.624 | — | 1.572 | 1.495 | 12.582 | 0.575 | **0.574** |
| CQA | Train $R^2$ | 0.666 | 0.534 | 0.677 | 0.323 | 0.139 | **0.890** | **0.890** |
| | Test MAE | 3.989 | 5.885 | 3.010 | 1697.096 | 5.483 | 2.203 | **2.182** |
| HumanEval | Train $R^2$ | 0.260 | — | 0.057 | 0.179 | **0.685** | 0.655 | 0.652 |
| | Test MAE | 2.889 | — | 2.389 | 3.341 | 2.111 | 2.041 | **2.025** |
| **Average Train $R^2$ (↑)** | | 0.304 | 0.290 | 0.375 | 0.194 | 0.485 | 0.867 | **0.874** |
| **Average Test MAE (↓)** | | 3.709 | 5.123 | 3.377 | 302.642 | 5.173 | 1.455 | **1.384** |
| **1B → 13B + SFT (Pre-to-Post)** | | | | | | | | |
| **Average Train $R^2$ (↑)** | | 0.329 | 0.302 | 0.517 | 0.257 | 0.413 | 0.820 | **0.846** |
| **Average Test MAE (↓)** | | 4.375 | 5.251 | 4.236 | 27.062 | 3.932 | 1.549 | **1.304** |
| **1B → 32B (Pre-to-Pre)** | | | | | | | | |
| **Average Train $R^2$ (↑)** | | 0.312 | 0.349 | 0.352 | 0.205 | 0.488 | 0.820 | **0.826** |
| **Average Test MAE (↓)** | | 19.785 | 5.165 | 3.546 | 21.833 | 3.276 | 1.540 | **1.481** |

Table 2: 1B→13B and 1B→32B performance across mathematics, science, engineering, common-sense, and coding benchmarks. Train fitting and test is done using 5-fold cross validation (§ 4.1). Best value across methods is **bolded**, and second best is underlined.

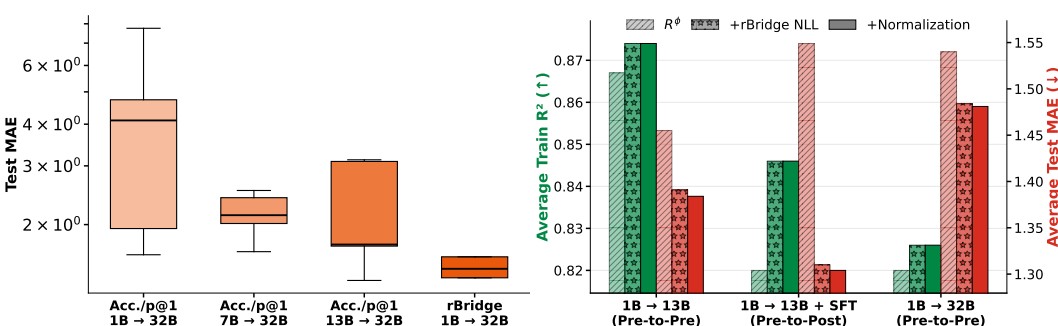

(a) RBRIDGE outperforms proxy models 7 - 13× larger using the target metric (Acc./p@1).

(b) Ablation shows that RBRIDGE NLL and normalization results in consistent improvement.

Figure 7: Additional experimental results demonstrate clear advantages of our proposed method.

sumption that small-scale performance meaningfully correlates with large-scale performance. Both works rely on standard NLL or perplexity metrics, which we demonstrate to be suboptimal in § 3.1. Furthermore, their experiments are predominantly on non-emergent benchmarks (e.g., TriviaQA and HellaSwag), whereas our work specifically targets reasoning benchmarks that exhibit emergent behavior. Lastly, their approaches are specialized for optimizing pre-training data *mixtures*, whereas our method addresses the more general problem of predicting large model performance with small proxy models, enabling *broader* applications.

Moreover, readers can view Prudêncio et al. (2024), Schellaert et al. (2025), Kipnis et al. (2025), and Pacchiardi et al. (2025) for some past works on LLM performance prediction, and evaluation. We acknowledge that alternative branches of the literature have studied predicting LLM performance.

| 1B → 7B Zero-shot Functional Relationship Transfer from $\mathcal{D}_{\text{pre}}$ to $\mathcal{D}'_{\text{pre}}$ | | | | | | | |
|---|---|---|---|---|---|---|---|
| **Method** | **Benchmark:** | **GSM8K** | **MATH500** | **ARC-C** | **MMLU Pro** | **CQA** | **Average** |
| **Ground Truth** | **Acc. (%) of $\mathcal{D}_{\text{pre}}$** | 10.538 | 2.800 | 56.911 | 10.225 | 60.442 | 28.183 |
| | **Acc. (%) of $\mathcal{D}'_{\text{pre}}$** | 8.264 | 3.600 | 51.536 | 9.578 | 44.554 | 23.506 |
| **$R^\phi$** | **Acc. (%) Prediction of $\mathcal{D}'_{\text{pre}}$** | 9.886 | 3.116 | 52.558 | 10.649 | 57.479 | 26.738 |
| | **Rank($\mathcal{D}_{\text{pre}}, \mathcal{D}'_{\text{pre}}$) w. Prediction (↑)** | ✓ | ✓ | ✓ | ✗ | ✓ | 4/5 |
| | **MAE (↓)** | 1.622 | 0.484 | 1.022 | 1.071 | 12.926 | 3.425 |
| **rBridge** | **Acc. (%) Prediction of $\mathcal{D}'_{\text{pre}}$** | 8.220 | 3.044 | 52.254 | 8.161 | 54.269 | 25.190 |
| | **Rank($\mathcal{D}_{\text{pre}}, \mathcal{D}'_{\text{pre}}$) w. Prediction (↑)** | ✓ | ✓ | ✓ | ✓ | ✓ | **5/5** |
| | **MAE (↓)** | 0.043 | 0.556 | 0.718 | 1.417 | 9.716 | **2.490** |

Table 3: RBRIDGE demonstrates strong zero-shot functional relationship transfer across datasets. Ground Truth rows are the Acc. performance and ranking ( rank 1 , rank 2 ) that the prediction is aiming to attain. RBRIDGE achieves 0 - 1.5 (%) error-rate excluding one outlier, and perfectly ranks the datasets on all five benchmarks. Average values are aggregated across the row.

**Minimal Compute Overhead.** While RBRIDGE introduces some additional costs, these are minor. First, is generating the gold reasoning trace $R^\phi$ via the frontier model. However, this represents a small one-time cost of under $10 per benchmark. To help the research community we plan to open-sourced our dataset, allowing future researchers to skip this cost for benchmarks we examine. Second, the automatic weighting mechanism (Fig. 1) incurs only a few seconds of CPU runtime per benchmark. Both overheads are negligible compared to the computational savings our method provides.

**RBRIDGE is a Better Predictor of Large Scale Pre-training Performance on Reasoning.** While we would expect the *same* metric to yield the greatest relationship and predictive power, our paper demonstrates that this is not the case. As visualized in Fig. 7a even as we scale the proxy model by $7\times$, $13\times$, it fails to achieve the performance levels of RBRIDGE. Our work empirically demonstrates that improved alignment with the **(1)** pre-training evaluation objective, and the **(2)** task is key to successfully leveraging small models to proxy large model's reasoning performance. Our ablative studies clearly indicate that each component **(1, 2)** of RBRIDGE is valuable (Tab. 1, Fig. 7b).

**Significant Compute and Economic Cost Reduction.** We observe definitive computational cost saving gains through RBRIDGE. In our first experiment **(i)**, we demonstrate that through RBRIDGE proxy models that are $324.3\times$ (proxy models $\frac{3.7}{1200}$ the size of target) smaller can be strong proxies for dataset ranking. From a computational perspective, RBRIDGE achieve at least $100.2\times$ compute savings against the best baseline in achieving the same ranking performance. In our second experiment **(ii)**, we demonstrate that $13\times$ and $32\times$ smaller proxy models can be effectively used as proxies. We believe that RBRIDGE's utility extrapolates to much greater proxy-target model scale differences. It is also important to note that, by extension, RBRIDGE has the potential to significantly reduce environmental footprint. This has meaningful societal impact as foundation model development has consequential environmental costs (Winsta, 2025; Morrison et al., 2025).

**Enabling Zero-shot Functional Relationship Transfer Across Datasets.** To our knowledge, we are the first to show that the proxy-target function fitted on one pre-training dataset (as shown in experiment **(ii)**) can be successfully transferred to an alternative one, with no additional fitting. We observe that the improved relationship fitting of RBRIDGE in experiment **(ii)**'s Tab. 2 extends to improved prediction (MAE) and ranking performance in Tab. 3. This saves $\frac{m}{n}\times$ compute as any additional pre-training dataset's performance can be reasonably approximated.

**Practical Potential Application: Two-stage Dataset Optimization.** Practitioners optimizing pre-training data navigate a vast candidate space of data sources and mixtures (Xie et al., 2023; Han et al., 2025; Liu et al., 2025). The experiments in **§ 4** provide a practically applicable framework for such dataset optimization that can be derived as a strong future work. **[Stage 1]** Given $N$ candidate datasets, following the setting of experiment **(i)** offers a cost-effective way to filter out $N - k$ poor datasets at e.g. <100M proxy scale, reducing the candidate space to $k$. With high decision accuracy of $\sim$80% (Fig. 6), we can expect bad datasets to have been filtered out correctly. Using relatively easier reasoning benchmarks (e.g., ARC-C) suffices for filtering out *weak* datasets.

This approach works because (1) strong candidates must also perform well on these tasks, and (2) their performance still correlates with more difficult ones (Fig. 12), even if they are not strong enough to serve as accurate predictors for ranking the downstream reasoning task. **[Stage 2]** For the remaining $k$ datasets, we train larger 1B-scale proxies following **(ii)** and **(iii)** to accurately rank them by predicting performance at the target scale (e.g., 32B). Although one training run at the target scale $m$ is needed, it is rare for practitioners to work without a baseline target model to improve upon.

Denoting the required proxy and target compute as $C_1^{\mathrm{p}}$ and $C_2^{\mathrm{p}}$ for proxies and $C^{\mathrm{t}}$ for target, it can be assumed that these costs $C$ are directly proportional to model size (Kaplan et al., 2020). The required total compute under this framework is $NC_1^{\mathrm{p}} + kC_2^{\mathrm{p}} + C^{\mathrm{t}}$ compared to the naïve approach cost $NC^{\mathrm{t}}$. Given that the cost reduction factor is $\frac{NC^{\mathrm{t}}}{NC_1^{\mathrm{p}}+kC_2^{\mathrm{p}}+C^{\mathrm{t}}}$, and $C_1^{\mathrm{p}}, C_2^{\mathrm{p}} \ll C^{\mathrm{t}}$, we can see how cost savings improve as the number of candidate datasets $N$ enlarge. Considering that the candidate space $N$ is very large in the real-world, this framework enables practitioners to optimize pre-training datasets at a fraction of the cost.

## 5.1 LIMITATION AND FUTURE DIRECTION

We discuss several limitations of RBRIDGE that present opportunities for future research. First, frontier models do not achieve perfect accuracy on reasoning tasks, potentially resulting in imperfect extracted reasoning trace $R^{\phi}$. While we utilize all available $R^{\phi}$ without filtering and demonstrate substantial performance and efficiency gains. Our preliminary experiments showed minimal improvements from post-hoc filtering of incorrect generations. Future work could investigate more sophisticated quality assurance mechanisms, such as ensemble methods leveraging multiple frontier models.

Second, frontier models occasionally fail to produce outputs in the required format for reasoning trace $R^{\phi}$ extraction. Our current approach provides one additional generation attempt before excluding the sample from training. This represents a limitation that future research could address through more robust prompt engineering, or ensembling multiple frontier models.

Finally, while we present a potential framework for further practical application in § 5, the efficient and effective implementation of this system remains an open challenge. Future work can further explore how to best leverage the methodological contribution of RBRIDGE (§ 3) and experimental robustness (§ 4) on more complex frameworks.

## 6 ETHICS STATEMENT

Our work conforms to the ICLR Code of Ethics. We report minor use of LLMs for polishing writing. All LLM-generated content has been reviewed and verified by the authors, who take full responsibility for the accuracy and integrity of the work presented.

## 7 REPRODUCIBILITY STATEMENT

Our method RBRIDGE is fully reproducible using the information provided in this paper. The approach follows a straightforward process: applying a specific prompt to a frontier model, then computing our proposed negative log-likelihood as presented in Eq. (1) and illustrated in Fig. 1. We provide the exact prompt in Appendix B along with pseudocode (Alg. 1) for implementation. To further support reproducibility and benefit the research community, we open-sourced our dataset.

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

# APPENDIX

## A ADDITIONAL RESULTS ON NOISY SMALL SCALE MODELS

Refer to Fig. 8.

## B RBRIDGE PSEUDOCODE

**Pseudocode.** The pseudocode is available in Alg. 1. We recommend viewing with Fig. 1 to build intuition. Recall that $x$ is an input from the benchmark dataset's validation or test we want to evaluate. Recall that $\pi^p, \pi^\phi$ are the proxy, and frontier LMs, respectively. $i$ denotes index of token $\tau$. $\tau^p$ denotes the token tokenized using $\pi^p$'s tokenizer. Exact prompt and extraction used in Line 1 is available in prompt.

---

**Algorithm 1: RBRIDGE**

**Input:** $x, \pi^p, \pi^\phi$

```
/* Step 1:  Extract reasoning trace and token-level confidence
   from frontier model                                          */
```
1   $R^\phi, p(R^\phi), A^\phi \leftarrow \pi^\phi(x)$ ;
```
/* Step 2:  Discard answer label, and compute normalized
   RBRIDGE NLL                                                  */
```
2   $w \leftarrow []$;

3   **for each** token $\tau_i^p$ **tokenizing** $R^\phi$ **do**

4     $w_i \leftarrow \text{Mean}_{\text{letter} \in \tau_i^p}(p^\phi(\text{letter}) \text{ from } p(R^\phi))$ ;

5     $w.\text{append}(w_i)$

6   $\text{RBRIDGE} \leftarrow []$;

7   **for each** token $\tau_i^p$ **tokenizing** $R^\phi$ **do**

8     $\text{NLL} \leftarrow -\log p^p(\tau_i^p) \sim \pi^p$ ;

9     $\text{RBRIDGE.append}(\text{NLL} \cdot \frac{w[i] - \min(w)}{\max(w) - \min(w)})$;

10   $\text{RBRIDGE} \leftarrow \text{Mean}(\text{RBRIDGE})$;

**Return:** RBRIDGE

---

**Prompt.** Here is the prompt we use to generate $R^\phi$. We use greedy decoding. Brackets [] indicate dynamic insertions depending on the benchmark and question. We extract the value for key "reasoning" to attain $R^\phi$.

> **Prompt to Frontier Model**
>
> System: You are a helpful assistant that solves [task] problems.
>
> User: [question]
> Respond ONLY with a JSON object in this exact format: { "reasoning": "your step by step reasoning", "final_answer": "your final answer" }

## C FURTHER EXPERIMENTAL DETAILS

**Common Evaluation Setting.** All evaluations are done using 5-shot CoT (Wei et al., 2022b).

### C.1 HARDWARE

We use A100 80G, H100 and H200 nodes and numerous terabytes of disk storage for experiments. For pre-training, we use 256 H100 GPUs with HBM3. For other hardware like CPU and RAM we use commonly available ones, as these hardware did not induce any bottlenecks.

## C.2 EXPERIMENT (I) ADDITIONAL DETAILS

All proxy models used is the default seed as provided in Magnusson et al. (2025). We do not use multi-seed averaging for the proxy as this significantly increases compute cost. We exactly follow Magnusson et al. (2025) on every other aspect of experiments, using their open-source assets where available. All intermediary checkpoints are as available from Magnusson et al. (2025).

## C.3 EXPERIMENT (II) ADDITIONAL DETAILS

**Additional Baseline Details.** iSFT results are unavailable for MMLU Pro (STEM) and HumanEval as these benchmarks do not provide an SFT set.

**Frontier Model's** $R^\phi$**.** Following Team (2025), we use `GPT 4o` to generate $R^\phi$. We tested `Claude 3.5 Sonnet` and `Gemini 2.5 Pro` and found no meaningful performance difference. We have not tested this method on long CoT models.

**Pre-training and Post-training Details.** Unless stated otherwise, for pre-training, we fully follow OLMo 2 (OLMo et al., 2024), and use their checkpoints where available. For SFT post-training we use the settings in Tab. 4.

Table 4: Hyperparameters for SFT

| Hyperparameter | Value |
|---|---|
| Epoch | 1 |
| Learning Rate | $1 \times 10^{-5}$ |
| Warmup Ratio | 0.1 |
| Batch Size | 64 |

## C.4 EXPERIMENT (III) ADDITIONAL DETAILS

**Alternative Pre-training Dataset** $\mathcal{D}'$**.** For the alternative pre-training dataset $\mathcal{D}'$ described in experiment **(iii)**, we closely follow the setup described in Han et al. (2025), but exclusively use publicly-available datasets. The training data follows an 8.5:1:0.5 ratio of English:multilingual:math/code, where the English portion consists of DCLM and FineWeb-Edu in equal proportions, and the multilingual portion comprises Korean, Chinese, and Japanese dumps of FineWeb and DCLM pipeline-processed Common Crawl filtered for Korean.

**Benchmark Choice.** We use the same benchmarks presented in **(ii)**, excluding HumanEval, as our extraction method on the alternative dataset $\mathcal{D}'$ achieved 0% p@1.

# D ADDITIONAL EXPERIMENTAL RESULTS

Refer to Fig. 9 and 10 for extended curve fitting visualization examples. Refer to Tab. 5 for all 1B → 13B (Pre-to-Post) benchmark results. MMLU Pro (STEM) and HumanEval are not included as they do not have a designated SFT set. Refer to Tab. 6 for all 1B → 32B (Pre-to-Pre) benchmark results. Refer to Fig. 11 for Kendall Tau results.

# E PROXY BENCHMARKS ARE UNRELIABLE

Refer to Fig. 12.

| Benchmark | Method: | Acc./p@1 | iSFT | TED | MPCA | NLL | $R^\phi$ | rBridge |
|---|---|---|---|---|---|---|---|---|
| GSM8K | Train R$^2$ | 0.363 | 0.385 | 0.565 | 0.129 | 0.868 | 0.951 | **0.956** |
| | Test MAE | 5.807 | 5.536 | 7.054 | 79.761 | 3.302 | 1.715 | **1.616** |
| MATH500 | Train R$^2$ | 0.088 | 0.128 | 0.199 | 0.182 | **0.557** | 0.544 | 0.555 |
| | Test MAE | 1.178 | 0.888 | 0.805 | 0.849 | 0.692 | 0.609 | **0.605** |
| ARC-C | Train R$^2$ | 0.187 | 0.153 | 0.522 | 0.331 | 0.160 | 0.938 | **0.964** |
| | Test MAE | 6.921 | 8.317 | 6.248 | 22.685 | 6.679 | 1.590 | **1.279** |
| CQA | Train R$^2$ | 0.677 | 0.543 | 0.783 | 0.385 | 0.068 | 0.845 | **0.910** |
| | Test MAE | 3.593 | 6.264 | 2.837 | 4.952 | 5.055 | 2.283 | **1.715** |
| **Average Train R$^2$ (↑)** | | 0.329 | 0.302 | 0.517 | 0.257 | 0.413 | 0.820 | **0.846** |
| **Average Test MAE (↓)** | | 4.375 | 5.251 | 4.236 | 27.062 | 3.932 | 1.549 | **1.304** |

Table 5: Detailed results of 1B → 13B+SFT (Pre-to-Post). Train fitting and test is done using 5-fold cross validation (§ 4.1). Best value across methods is **bolded**, and second best is underlined.

| Benchmark | Method: | Acc./p@1 | iSFT | TED | MPCA | NLL | $R^\phi$ | rBridge |
|---|---|---|---|---|---|---|---|---|
| GSM8K | Train R$^2$ | 0.281 | 0.398 | 0.706 | 0.076 | 0.814 | **0.972** | **0.972** |
| | Test MAE | 7.755 | 7.299 | 6.505 | 107.493 | 6.935 | 1.676 | **1.601** |
| MATH500 | Train R$^2$ | 0.193 | 0.196 | 0.166 | 0.019 | 0.823 | 0.832 | **0.834** |
| | Test MAE | 98.093 | 1.869 | 1.985 | 6.907 | 0.815 | 0.739 | **0.718** |
| ARC-C | Train R$^2$ | 0.206 | 0.169 | 0.582 | 0.346 | 0.144 | 0.926 | **0.949** |
| | Test MAE | 4.734 | 5.976 | 4.491 | 7.588 | 4.985 | 1.560 | **1.384** |
| MMLU Pro (STEM) | Train R$^2$ | 0.057 | — | 0.031 | 0.210 | 0.344 | **0.756** | **0.756** |
| | Test MAE | 2.075 | — | 1.930 | 1.864 | 2.133 | **1.326** | **1.326** |
| CQA | Train R$^2$ | 0.624 | 0.634 | 0.528 | 0.332 | 0.046 | 0.765 | **0.766** |
| | Test MAE | 4.105 | 5.516 | 3.630 | 4.199 | 3.664 | **2.382** | 2.384 |
| HumanEval | Train R$^2$ | 0.514 | — | 0.101 | 0.245 | **0.759** | 0.666 | 0.679 |
| | Test MAE | 1.946 | — | 2.733 | 2.947 | **1.424** | 1.555 | 1.474 |
| **Average Train R$^2$ (↑)** | | 0.312 | 0.349 | 0.352 | 0.205 | 0.488 | 0.820 | **0.826** |
| **Average Test MAE (↓)** | | 19.785 | 5.165 | 3.546 | 21.833 | 3.276 | 1.540 | **1.481** |

Table 6: Detailed results of 1B → 32B (Pre-to-Pre). Train fitting and test is done using 5-fold cross validation (§ 4.1). Best value across methods is **bolded**, and second best is underlined.

## F    ATTRIBUTION

We attribute the neural network icon in Fig. 1 as taken from Freepik (flaticon.com). Their guideline indicates that it can be used with attribution.

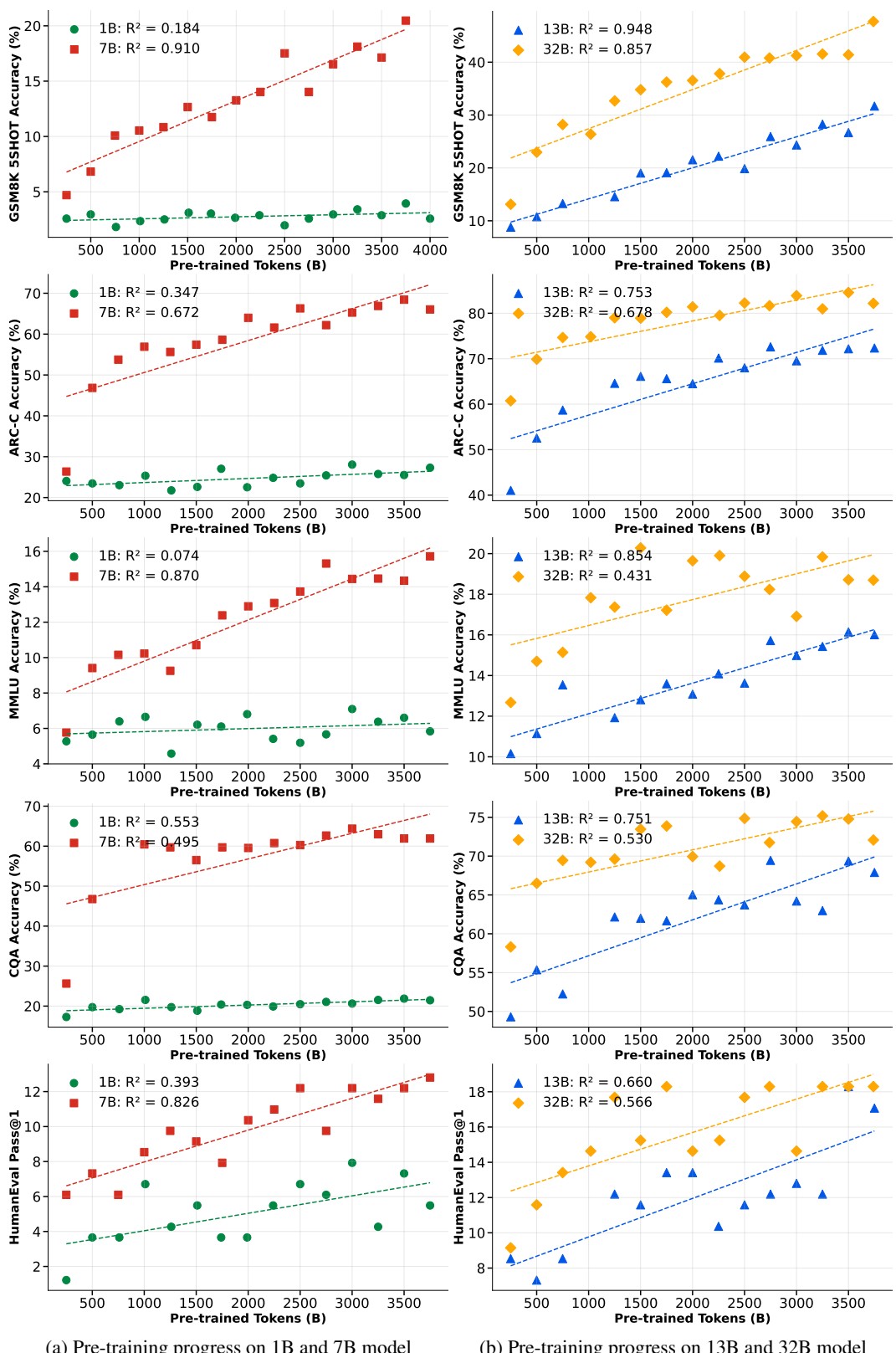

(a) Pre-training progress on 1B and 7B model

(b) Pre-training progress on 13B and 32B model

Figure 8: Given the same data source, smaller models exhibit significantly more noise and occasionally provide the wrong direction, making it challenging to use smaller models to proxy larger model performance. $R^2$ values are derived from linear curve fitting. MMLU corresponds to MMLU Pro (STEM).

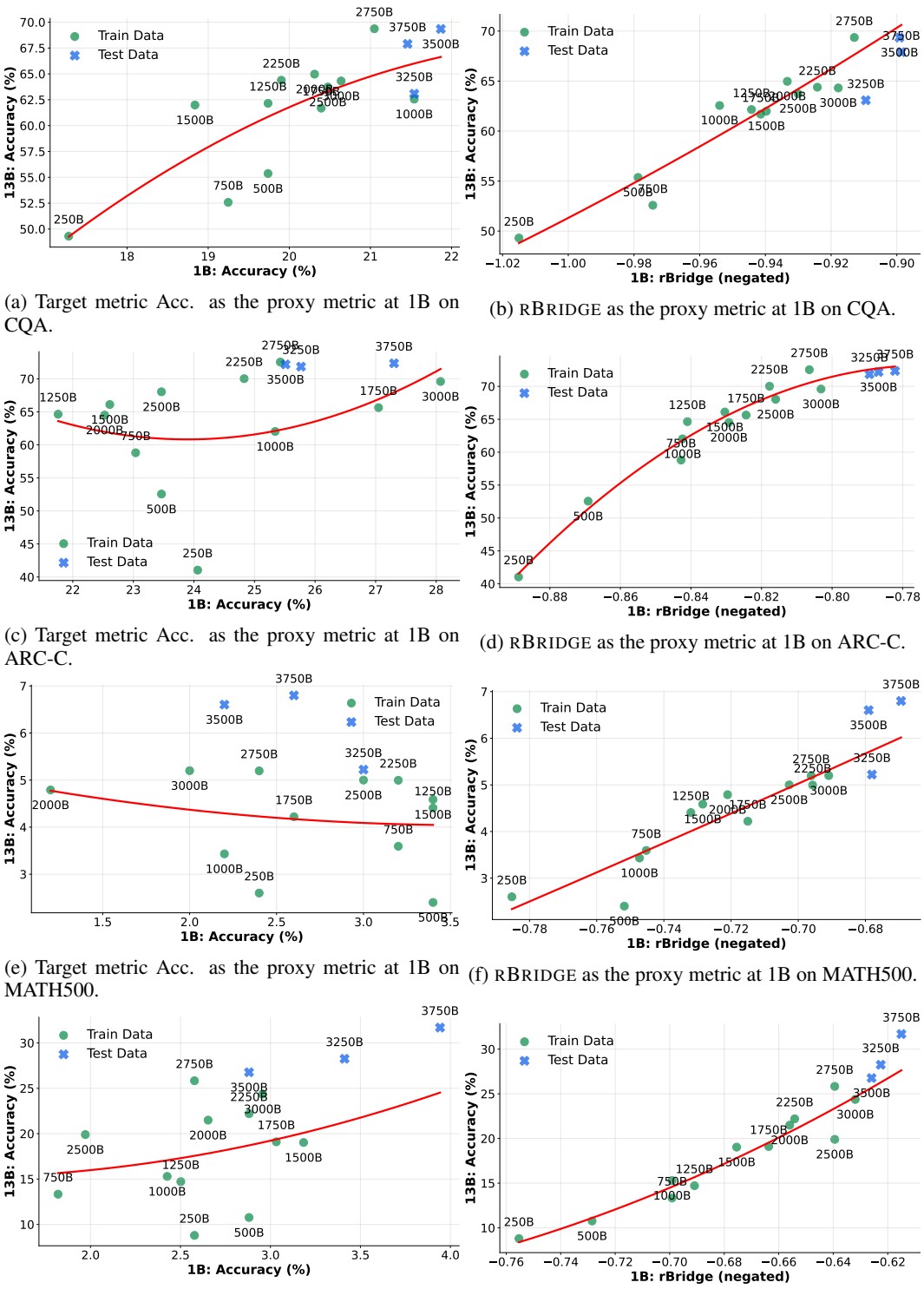

(a) Target metric Acc. as the proxy metric at 1B on CQA.

(b) RBRIDGE as the proxy metric at 1B on CQA.

(c) Target metric Acc. as the proxy metric at 1B on ARC-C.

(d) RBRIDGE as the proxy metric at 1B on ARC-C.

(e) Target metric Acc. as the proxy metric at 1B on MATH500.

(f) RBRIDGE as the proxy metric at 1B on MATH500.

(g) Target metric Acc. as the proxy metric at 1B on GSM8K.

(h) RBRIDGE as the proxy metric at 1B on GSM8K.

Figure 9: Example visualization from a fold of proxy-target relationship study at 1B → 13B on CQA, ARC-C, MATH500, and GSM8K. Each data point represents equal trained tokens for the proxy and target model.

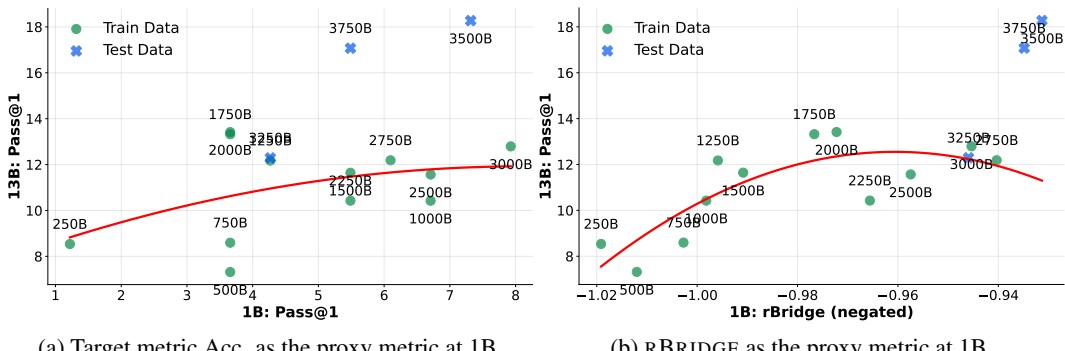

(a) Target metric Acc. as the proxy metric at 1B.

(b) RBRIDGE as the proxy metric at 1B.

Figure 10: Example visualization from a fold of proxy-target relationship study at 1B → 13B on HumanEval. Each data point represents equal trained tokens for the proxy and target model.

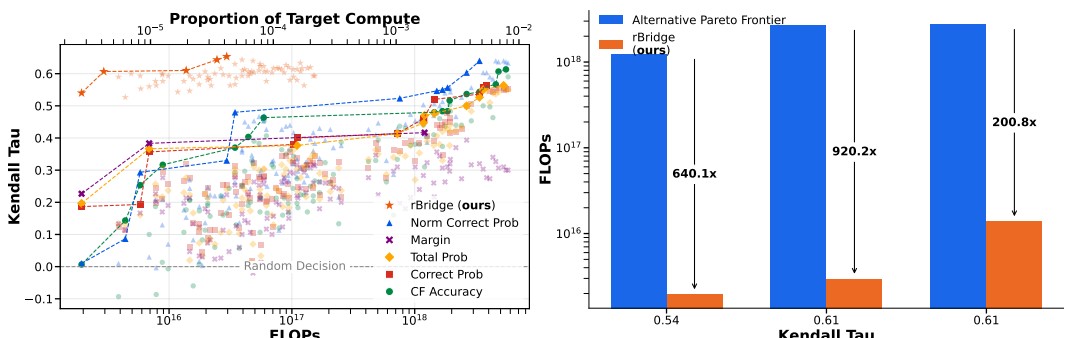

(a) Kendal Tau results across 3.7 - 97.9M proxy model size, and across intermediary checkpoints.

(b) RBRIDGE saves FLOPs by a factor of 200.8× to 920.2×.

Figure 11: RBRIDGE improves the pareto frontier in pre-training dataset ranking for 1.2B target model size. Values are averages aggregated across ARC-C and CQA. The two most compute efficient points in RBRIDGE's pareto frontier is (1) 3.7M model size trained on 87.3M tokens, and (2) 6M model size trained on 81.6M tokens.

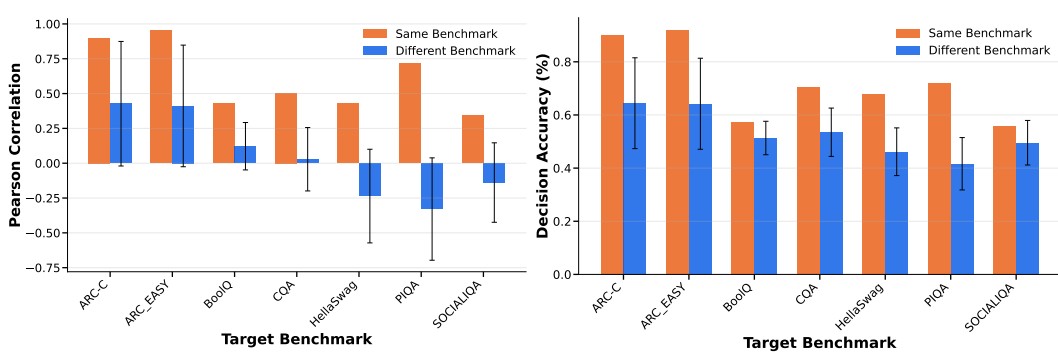

Figure 12: Correlation and decision accuracy performance using the same benchmark vs. different benchmark on 97.9M → 1.2B. Error bars indicate one standard deviation. On average, different benchmarks performs sub-optimally and are noisy.

