# OpenReview forum: "Predicting LLM Reasoning Performance with Small Proxy Model"
_ICLR.cc/2026/Conference — ICLR 2026 Poster_

### Official Review · Reviewer_KWnN · 2025-10-24

**Soundness:** 3
**Presentation:** 2
**Contribution:** 4
**Rating:** 8
**Confidence:** 3

**Summary:**

This paper offers an improved method for predicting large language model performance on reasoning tasks using results from smaller proxy models. RBridge is an elegant solution. First, reasoning traces from a frontier model are extracted for some benchmark dataset. Second, the negative log-likelihood of each token in the reasoning trace is computed using a small proxy language model. Finally, this token-wise NLL is weighted by MinMax normed average probability of each character in that token, as defined by the frontier model. This produces a weighted negative log-likelihood of the reasoning trace. The authors show that this value ranks and predicts performance of larger models.

In the first experiment, they train small proxy models and large target-size models on 25 different datasets. They compute the agreement (Decision Accuracy) between the rankings of which dataset leads to better performance (on specific benchmarks) for the proxy models and for the target models. They show that they outperform other methods for ranking datasets and incur a significant compute saving.

In the second experiment, they show that the RBridge value (weighted NLL) for proxy models at different stages of training is strongly correlated with and predictive of the performance of much larger models.

In the third experiment, they show that fitting a function to predict accuracy given the RBridge value (weighted NLL) based on one dataset, transfers seamlessly to predicting performance for a model trained on an entirely different dataset.

Overall, this work offers an innovative and powerful new metric for optimising pre-training datasets through the use of small proxy models. This will be very useful to the community for improving LLM pre-training efficiently, because they can iterate on different training regimes using a small model, with the knowledge that they can predict the performance of much larger models.

**Strengths:**

The paper has many strengths:
* The motivation and methodology are well thought through and sound.
* The empirical work is diligent and complete. I commend the authors on the number of training and testing runs they must have performed to complete this project.
* The results are very impressive. The efficiency gain from using RBridge compared to other methods is considerable and will greatly accelerate the development of optimal pre-training datasets for large-scale reasoning models.

**Weaknesses:**

There are only a few minor weaknesses with the paper at the moment:
1. The authors miss some important literature on predictable AI and the use of assessor models for predicting model performance on unseen tasks. I include relevant references at the bottom.
2. The writing is quite rough in several areas. I outline the key spelling/syntax problems below. I would recommend going over the whole paper to check for sense and grammaticality.
3. I would include the limitations and future directions section in the main text, given the extra page.
4. The authors are right to point out that obtaining the reasoning traces from frontier models is an extra cost and may be imperfect. I would definitely like to see the authors make their datasets public upon publication.
5. This is not really a weakness, but the authors could emphasise more that their results really mean that LLM development can *greener*. Right now, iterating over different data regimes uses a lot of energy, which their method can help to reduce if the community makes use of RBridge.


## Missing References

Kipnis, A., Voudouris, K., Buschoff, L. M. S., & Schulz, E. (2024). metabench--A Sparse Benchmark of Reasoning and Knowledge in Large Language Models. arXiv preprint arXiv:2407.12844.

Pacchiardi, L., Voudouris, K., Slater, B., Martínez-Plumed, F., Hernández-Orallo, J., Zhou, L., & Schellaert, W. (2025). PredictaBoard: Benchmarking LLM score predictability. arXiv preprint arXiv:2502.14445.

Prudêncio, R. B., Lorena, A. C., Silva-Filho, T., Drapal, P., & Valeriano, M. G. (2024, June). Assessor models for explaining instance hardness in classification problems. In 2024 International Joint Conference on Neural Networks (IJCNN) (pp. 1-8). IEEE.

Schellaert, W., Martínez-Plumed, F., & Hernández-Orallo, J. (2025). Analysing the predictability of language model performance. ACM Transactions on Intelligent Systems and Technology, 16(2), 1-26.


## Spelling/Syntax

Approximation line numbers followed by the corrected sentence.

Line 36: "suggests that there may be a limit..."
Line 88: "To bridge the evaluation scheme between small proxy models to large target-scale models..."
Line 94: "We improve task alignment at ...."
Line 95 and the enumerated list: This needs to be fully revised for sense. It should start with "We empirically validate our method in the following five ways:" and then each number should be a full sentence.
Line 269: "which demonstrates that including an SFT stage..."
Line 286: "can help target metric Acc./p@1 to be used as signals"
Line 286-7: The sentence on TED oes not make sense to me.
Line 289: "Last is ScB which visualizes..."
Line 299: "respectively"

**Questions:**

* Are letter-level probabilities within each token conditional on the previous characters/tokens?
* The performance of $R^\phi$ alone is surprisingly high (e.g., in Table 2). Why bother with the extra hassle of character-wise tokenization etc.? What argument can the authors make about the meaningfulness of the difference in performance between $R^\phi$ and RBridge? To some, these differences might look negligible.

---

> ### Author Response · Authors · 2025-11-19
> **Response to Reviewer KWnN (Part I)**
>
> Thank you for taking your time reviewing our paper. We have found your feedback highly constructive, and would like to address your concerns **[W]** and questions **[Q]** below. We are grateful for your feedback, which has helped us improve our paper.
>
> ### **[W1] Added Additional References**
>
> In response to your comment:
> > The authors miss some important literature on predictable AI and the use of assessor models for predicting model performance on unseen tasks. I include relevant references at the bottom.
>
> We have added these references to **Sec. 5 Additional Related Work**.
> > Moreover, readers can view [1], [2], [3], and [4] for some past works on LLM performance prediction, and evaluation. We acknowledge that alternative branches of the literature have studied predicting LLM performance.
>
> ### **[W2] Fixed Minor Spelling and Syntax Suggestions**
>
> In response to your comment:
> > The writing is quite rough in several areas. I outline the key spelling/syntax problems below. I would recommend going over the whole paper to check for sense and grammaticality.
>
> Spelling and syntax errors have been corrected in the revised manuscript:
>
> > suggest $\rightarrow$ suggests
>
> > further $\rightarrow$ improve
>
> > demonstrate $\rightarrow$ demonstrates
>
> > help $\rightarrow$ helps
>
> > visualize $\rightarrow$ visualizes
>
> > repsectively $\rightarrow$ respectively
>
> > Third is Token Edit Distance (TED; \cite{NEURIPS2023_adc98a26}) which argue that emergence occurs due to discontinuous metrics.
> $\rightarrow$
> Third is Token Edit Distance (TED; \cite{NEURIPS2023_adc98a26}) which argue that emergence occurs due to discontinuous metrics, and correspondingly proposes their continuous metric.
>
> Our **Sec. 1 Introduction Contribution**'s numerical bullet points have been revised to full sentences for clarity. These can be checked in the updated PDF version with revised text highlighted in **$\textcolor{blue}{blue}$** for convenience.
>
> ### **[W3] Added Limitation and Future Directions to Main Body**
>
> In response to your comment:
> > I would include the limitations and future directions section in the main text, given the extra page.
>
> We have moved the limitation and future direction to the main body in the updated version.
>
> ### **[W4] Commited to Open Source and Open Data**
>
> In response to your comment:
> > The authors are right to point out that obtaining the reasoning traces from frontier models is an extra cost and may be imperfect. I would definitely like to see the authors make their datasets public upon publication.
>
> We are committed to easy reproducibility and will release both code (on GitHub) and datasets (on HuggingFace) upon acceptance. This includes all reasoning traces extracted from frontier models, enabling the community to build upon our work without incurring additional extraction costs. Nevertheless, for the time being we believe that our work is fully reproducible with intuitive visualization (**Fig. 1**) and a straight-forward pseudocode (**Appendix B**, **Alg. 1**) with the prompt explicitly stated.
>
> ### **[W5] We Agree that our Method Helps The Environment**
>
> In response to your comment:
> > This is not really a weakness, but the authors could emphasise more that their results really mean that LLM development can greener. Right now, iterating over different data regimes uses a lot of energy, which their method can help to reduce if the community makes use of RBridge.
>
> You raise an excellent point that our method significantly helps reduce environmental footprint as less compute can be used in the pre-training stage. To highlight this, we have included the following part in the revised version's **Sec. 5 Significant Compute and Economic Cost Reduction.**:
> > It is also important to note that, by extension, \tsc{rBridge} has the potential to significantly reduce environmental footprint. This has meaningful societal impact as foundation model development has consequential environmental costs [5], [6].

---

> ### Author Response · Authors · 2025-11-19
> **Response to Reviewer KWnN (Part II)**
>
> ### **[Q1] Letter-level Probabilities are Conditioned on Previous Tokens**
>
> Yes, all probabilities are conditioned on the context, following standard auto-regressive transformer decoders.
>
> ### **[Q2] $R^\phi$ $\rightarrow$ rBridge Performance is Not Negligable**
>
> While the numerical difference in **Tab. 2** appear modest, **Tab. 3** demonstrates their practical significance: rBridge achieves perfect dataset ranking (5/5 correct ranking) while $R^\phi$ alone achieves only 4/5, despite seemingly small MAE differences. In pre-training dataset selection, where a single incorrect ranking decision can waste *thousands* of GPU-hours, even marginal predictive improvements are crucial.
>
> ### **Reference**
>
> [1] Prudencio et al. Assessor Models for Explaining Instance Hardness in Classification Problems. IJCNN 2024
>
> [2] Kipnis et al. metabench - A Sparse Benchmark of Reasoning and Knowledge in Large Language Models. ICLR 2025
>
> [3] Pacchiardi et al. PredictaBoard: Benchmarking LLM Score Predictability. ACL 2025 Findings
>
> [4] Schellaert et al. Analysing the Predictability of Language Model Performance. ACM Transactions on Intelligent Systems and Technology 2025.
>
> [5] Morrison et al. Holistically Evaluating the Environmental Impact of Creating Language Models. ICLR 2025 Spotlight
>
> [6] Winsta. The Hidden Costs of AI: A Review of Energy, E-Waste, and Inequality in Model Development. 2025

---

> > ### Comment · Reviewer_KWnN · 2025-11-19
> >
> > Thanks for your prompt response. I am happy with the revisions and will keep my score.

---

> ### Author Response · Authors · 2025-11-20
>
> Thank you for helping us improve our paper.
>
> We hope that our discussion has helped increase your confidence in our work.
>
> Please let us know if you have any other inquiries, or points to improve on.
>
> Best regard,
> Authors

---

### Official Review · Reviewer_v8sd · 2025-10-30

**Soundness:** 3
**Presentation:** 4
**Contribution:** 3
**Rating:** 6
**Confidence:** 4

**Summary:**

This paper proposes to reliably estimate the reasoning capabilities of large LLMs using small proxy models.
The paper shows that, as the reasoning capability is an emergent property of large LLMs, the vanilla loss obtained from small models can be noisy.
There are two key limitations identified: (i) evaluation objective misalignment (e.g., NLL loss v.s. pass@K, and off-policy Y distribution ) (ii) task alignment and at the token-level (e.g., important tokens like branching/searching/backtracking should be taken into consideration)

The proposed rBridge tackles the limitations via (i) using a reasoning trace generated by frontier models as a proxy of Y and (ii) the loss is weighted according to the normalized log-probs of the frontier model, where the potential tokenization mismatch issue is tackled by using the character-level average.

Experimental results show the proposed rBridge successfully predicts the model performance using much smaller proxy models (< 100M) for a 1.2B target model. The correlation stays strong when using the 1B model to predict 13B/32B models and shows promising transferability across pre-training datasets.

**Strengths:**

- The investigated problem is of great practical value, reducing the computational cost for training large-scale LLMs.
- The rBridge is generally effective and easy to implement, with details well covered.

**Weaknesses:**

- While the papers examine the choices of frontier models in the appendix (i.e., no meaningful difference), I am still concerned about whether the quality of the reasoning trajectories would influence the estimation. As frontier models remain a black box, it is challenging to understand the underlying training distribution. Would the prediction performance degrade significantly if the reasoning benchmarks are OD even for those frontier models?
- The weighted NLL is not ablated. How much does it contribute to the final prediction performance?

**Questions:**

See Weaknesses

---

> ### Author Response · Authors · 2025-11-19
> **Response to Reviewer v8sd**
>
> Thank you for taking your time reviewing our paper. We would like to address your concerns **[W]** and questions **[Q]** below. Your contribution has helped us improve the quality of our paper.
>
> ### **[W1] Pre-training Benchmarking on Tasks that Frontier Models Fails At is Lower Priority**
>
> In response to your comment:
> > While the papers examine the choices of frontier models in the appendix (i.e., no meaningful difference), I am still concerned about whether the quality of the reasoning trajectories would influence the estimation. As frontier models remain a black box, it is challenging to understand the underlying training distribution. Would the prediction performance degrade significantly if the reasoning benchmarks are OD even for those frontier models?
>
> As you mentioned, while we found no meaningful difference across frontier models:
> > Following [1], we use \ttt{GPT 4o} to generate $R^\phi$. We tested \ttt{Claude 3.5 Sonnet} and \ttt{Gemini 2.5 Pro} and found no meaningful performance difference.
>
> , we acknowledge your point on benchmarks that remain challenging even for frontier models.
>
> We respectfully argue that pre-training optimization should be evaluated differently from later training stages. Modern LLM development follows a multi-stage pipeline [2,3,4,5]:
> > Pre-training → Mid-training → SFT → RLVR → Prompting and inference-time methods
>
> For pre-training specifically, we contend that using benchmarks where frontier models already perform reasonably well (e.g., MMLU Pro, GPQA Diamond) is methodologically appropriate because pre-training establishes foundational capabilities that are refined in subsequent stages. Capabilities on *extremely* challenging benchmarks can be further developed through mid-training, SFT, RLVR, and inference-time methods.
>
> While we believe that this is not a substantive weakness, we agree that this is a valuable future empirical study. One promising extension would be to leverage inference-time scaling methods (e.g., Pass@K with large K, or multi-agent systems) to obtain high-quality reasoning traces even on benchmarks where frontier models struggle on. In principle, this would enable rBridge to work effectively on more challenging tasks.
>
> ### **[W2] Ablation Study of Weighted NLL Already Exists**
>
> In response to your comment:
> > The weighted NLL is not ablated. How much does it contribute to the final prediction performance?
>
> Please consider **Fig. 7 (b)** where we have ablated the weighted NLL. Starting from $R^\phi$, there is a meaningful performance gain using rBridge NLL. Next, if we normalize the weighting (which amplifies its effect), we see additional consistent predictive gains across all three proxy settings. We discussed this in the main text as well in line 355 - 358.
>
> Please also consider **Tab. 3**, which empirically analyzes going from $R^\phi \rightarrow$ rBridge; which is essentially post-NLL weighting. There is a clear performance gain from $R^\phi \rightarrow$ rBridge, where rBridge is able to achieve perfect 5/5 correct dataset ranking, while $R^\phi$ only achieves 4/5.
>
> ### **Reference**
> [1] MiniCPM Team. MiniCPM4: Ultra-Efficient LLMs on End Devices. 2025
>
> [2] Qwen Team. Qwen2.5 Technical Report. 2025
>
> [3] Qwen Team. Qwen3 Technical Report. 2025
>
> [4] Qwen Team. Qwen2.5-VL Technical Report. 2025
>
> [5] DeepSeek-AI. DeepSeek-R1: Incentivizing Reasoning Capability in LLMs via Reinforcement Learning. 2025

---

> > ### Comment · Reviewer_v8sd · 2025-11-28
> >
> > Thanks for your clarification. I keep my original rating.

---

### Official Review · Reviewer_bXEE · 2025-10-30

**Soundness:** 3
**Presentation:** 3
**Contribution:** 3
**Rating:** 4
**Confidence:** 4

**Summary:**

This paper introduces RBRIDGE, a method for predicting large language model reasoning performance using small proxy models (≤1B parameters) by aligning evaluation with both the pre-training objective and the target task. RBRIDGE leverages reasoning traces from frontier models as gold labels and introduces a weighted negative log-likelihood metric that emphasizes task-critical tokens, achieving strong correlations across six reasoning benchmarks and reducing dataset ranking costs by over 100×. It further demonstrates zero-shot transfer of predictive relationships across pre-training datasets, enabling efficient performance estimation at large scale.

**Strengths:**

1. The discussed problem is both intellectually engaging and practically important for advancing efficient LLM development.

2. The proposed RBRIDGE metric shows strong performance, achieving high correlation with large-model reasoning while greatly reducing computational costs.

**Weaknesses:**

1. The paper shows small models struggle with noisy tasks but doesn't clearly explain how to identify which datasets are noisy—more analysis on dataset characteristics would help.

2. The frontier model choice matters: models like R1 and GPT-4o differ in style and training data, which could affect reasoning trace quality and RBRIDGE’s consistency.

3. RBRIDGE is tested on math and QA tasks, but performance may differ on code (e.g., LiveCodeBench) or subjective tasks (e.g., Arena-Hard), where reasoning patterns vary.

4. Figure 5(b) illustrates RBRIDGE’s strong fit, but similar plots on other benchmarks would strengthen the empirical support.

5. Minor typos need fixing, e.g., “is becomes” in Figure 1’s caption.

6. The paper exceeds 9 pages, which violates the rule.

**Questions:**

See weakness.

---

> ### Author Response · Authors · 2025-11-19
> **Response to Reviewer bXEE (Part I)**
>
> Thank you for taking your time reviewing our paper. We would like to address your concerns **[W]** and questions **[Q]** below. Your contribution has helped us improve the quality of our paper.
>
> ### **[W1] We Have Analyzed Why Some Benchmark Datasets are Noisy at Small Model Scale**
>
> In response to your comment:
> > The paper shows small models struggle with noisy tasks but doesn't clearly explain how to identify which datasets are noisy—more analysis on dataset characteristics would help.
>
> We analyze why different benchmarks exhibit different levels of noise when using small model proxies (**Fig. 3**). Our key finding is that benchmarks with gold labels $Y^\*$ with higher NLL under the small model exhibit significantly noisier proxy performance. This connects dataset noisiness directly to distributional mismatch between $Y^\*$ and the pre-training data. We empirically validate this in **Tab. 1**, which shows that more in-distribution (ID) gold label $Y^\*$ is directly related to improved correlation.
>
> We explain why some gold label $Y^\*$ are more out-of-distribution (OOD) in **Sec. 3.1 (1.b) Not All NLL are Equal: Distributional Alignment**. As seen by the qualitative example in line 191 - 192, parts of gold label $Y^\*$ that do not commonly occur in the pre-training corpora lead to higher NLL (more OOD). In the case of MMLU Pro (STEM), ARC-C, and CQA which exhibit high NLL, this is primarily due to the multiple-choice answer strings (e.g., ‘Answer: (A)…’) that rarely occur in the pre-training corpus, leading to elevated NLL and thus noisier predictions from small models. For instance MATH500 and GSM8K embody lower NLL as their gold label $Y^\*$ is not multiple choice question format.
>
> We have further discussed this in **Sec. 3.2**, where we analyze how the gold labels $Y^\*$ we generate $R^\phi$'s close distribution with the pre-training data yields the best predictive results.
>
> ### **[W2] Our Method is Frontier Model Choice Agnostic**
>
> In response to your comment:
> > The frontier model choice matters: models like R1 and GPT-4o differ in style and training data, which could affect reasoning trace quality and RBRIDGE’s consistency.
>
> Our method is not GPT 4o-specific: **(i)** we have discussed how preliminary experiments with other frontier models showed no meaningful performance difference, **(ii)** it was important to follow our baseline [1] for replicability and fairness, and **(iii)** the paper's findings also help practioners and researchers choose frontier model.
>
> **(i)** We discuss frontier model choice in **Appendix C.3**.
>
> > Following [1], we use GPT 4o to generate $R^\phi$. We tested Claude 3.5 Sonnet and Gemini 2.5 Pro and found no meaningful performance difference. We have not tested this method on long CoT models.
>
> While we have not yet evaluated long-CoT frontier models such as DeepSeek-R1, our method depends primarily on the distribution of the reasoning trace, not on length. Evaluating richer long-CoT models is an excellent direction for future work but does not affect the core empirical finding that frontier reasoning traces improve predictability.
>
> Moreover, on the concern of quality, we have discussed this in **Appendix F**:
> > First, frontier models do not achieve perfect accuracy on reasoning tasks, potentially resulting in imperfect extracted reasoning trace $R^\phi$. While we utilize all available $R^\phi$ without filtering and demonstrate substantial performance and efficiency gains. Our preliminary experiments showed minimal improvements from post-hoc filtering of incorrect generations. Future work could investigate more sophisticated quality assurance mechanisms, such as ensemble methods leveraging multiple frontier models.
>
>
> **(ii)** It was especially reasonable to use GPT 4o following ScB (our baseline) [1] ensuring that our replication and comparison is fair.
>
> **(iii)** Our findings also provide a practical diagnostic: practitioners can compute the NLL of candidate frontier-model reasoning traces to select the model whose traces correspond to the lowest NLL with the small model. This is backed by the our empirical analysis in **Tab. 1**.

---

> ### Author Response · Authors · 2025-11-19
> **Response to Reviewer bXEE (Part II)**
>
> ### **[W3] Coding Reasoning Benchmark is Not Missing; and Subjective Benchmarks are Out-of-scope**
>
> In response to your comment:
> > RBRIDGE is tested on math and QA tasks, but performance may differ on code (e.g., LiveCodeBench) or subjective tasks (e.g., Arena-Hard), where reasoning patterns vary.
>
> Our evaluation already includes HumanEval, a widely used coding reasoning benchmark, covering algorithmic reasoning and program synthesis. Thus, coding tasks are represented in our study. We believe that our empirical study is sufficiently comprehensive. Consider the benchmarks we use:
>
> | Benchmark | Reasoning Task |
> | -------- | -------- |
> | GSM8K     | Math     |
> | MATH500     | Math     |
> | ARC-C | Science |
> | CQA | Commonsense |
> | MMLU Pro (STEM) | Science, Computer Science, Engineering, Math |
> | HumanEval | Coding |
>
> Subjective or preference-aligned benchmarks (e.g., Arena-Hard) measure value judgments rather than reasoning. Because our work focuses specifically on verifiable reasoning tasks, such benchmarks fall outside our scope, similar to prior influential reasoning papers:
>
> [2] ReGenesis (ICLR 2025 Oral): Math, Logical, Commonsense
>
> [3] STaR (NeurIPS 2022, 1183 cites): Math, Commonsense
>
> [4] Self-Consistency (ICLR 2023, 3224 cites): Math, Science, Commonsense
>
> ### **[W4] Added Additional Curve Fitting Example Visualizations**
>
> In response to your comment:
> > Figure 5(b) illustrates RBRIDGE’s strong fit, but similar plots on other benchmarks would strengthen the empirical support.
>
> We have now added curve-fitting visualizations for all evaluated benchmarks in **Appendix D**, complementing our exsiting quantitative metrics R$^2$, MAE available in **Tab. 2** and **3**.
>
> We have added
> > Extended visualizations for all other benchmarks available in Appendix D.
>
> in the caption of **Fig. 5**.
>
> ### **[W5] Fixed Minor Typos**
>
> In response to your comment:
> > Minor typos need fixing, e.g., “is becomes” in Figure 1’s caption.
>
> We appreciate your feedback, and have accordingly updated
> > how in-distribution the target Y$^∗$ is becomes important
>
> to
>
> > whether the target $Y^\*$ is in-distribution becomes important.
>
> ### **[W6] Paper Does Not Exceed 9 Pages (Ethics/Reproducibility Statement Does Not Count)**
>
> In response to your comment:
> > The paper exceeds 9 pages, which violates the rule.
>
> At the time of original submission (pre-review) our page limit did *not* exceed 9 pages. **Sec. 6 Ethics Statement**, and **Sec. 7 Reproducibility Statement** that came in page 10 (at the time of submission) are not included in the page limit by ICLR policy:
> > The optional ethics statement will not count toward the page limit, but should not be more than 1 page.
>
> and
>
> > This optional reproducibility statement is not part of the main text and therefore will not count toward the page limit.
>
> taken directly from the ICLR 2026 Author Guide. Our revised version during the review and rebuttal phase may exceeed 9 pages as this is allowed:
> > During the discussion/rebuttal phase and for the camera ready, the page limit will be increased to 10 pages to allow for new results/discussions.
>
> ### **Reference**
>
> [1] MiniCPM Team. MiniCPM4: Ultra-Efficient LLMs on End Devices. 2025
>
> [2] Peng et al., SalesForce AI Research. ReGenesis: LLMs can Grow into Reasoning Generalists via Self-Improvement. ICLR 2025
>
> [3] Zelikman et al. STaR: Bootstrapping Reasoning With Reasoning. NeurIPS 2022
>
> [4] Wang et al. Self-Consistency Improves Chain of Thought Reasoning in Language Models. ICLR 2023

---

### Official Review · Reviewer_4B33 · 2025-11-01

**Soundness:** 3
**Presentation:** 2
**Contribution:** 2
**Rating:** 4
**Confidence:** 2

**Summary:**

This paper introduces a new method to use a small proxy model to predict the reasoning performance of LLM. It uncovers that existing methods don’t work well because they fail to align with the pre-training objective and the target task. To address the limitations, the proposed method uses frontier-model generated gold reasoning traces for better alignment. Experimental results demonstrate the effectiveness of the proposed method.

**Strengths:**

+ using frontier-model generated gold reasoning traces for better alignment
+ significantly improved performance in terms of compute cost, proxy model size, and zero-shot transferring from one pre-trained dataset to another

**Weaknesses:**

- needs a frontier-model to generate gold reasoning traces, making the proposed method less useful
- it is unclear how the small proxy models less than <100M work so well

**Questions:**

How was the compute cost measured? Is it measured in a real system or just calculated by the model size? What are the system (hardware and software) settings if a real system was implemented? Using the proposed method, can a larger proxy model work better and how much better than a smaller proxy model?

---

> ### Author Response · Authors · 2025-11-18
> **Response to Reviewer 4B33 (Part I)**
>
> Thank you for taking your time reviewing our paper. We would like to address your concerns **[W]** and questions **[Q]** below. Your contribution has helped us improve the quality of our paper.
>
> ### **[W1] Leveraging an Existing Frontier-model to Advance Next Generation Foundation Model Training is Common Practice**
>
> In response to your comment:
> > needs a frontier-model to generate gold reasoning traces, making the proposed method less useful
>
> This design choice is *both practical and aligned with standard practice* in modern foundation model development [1, 2, 3, 4, 5]. We respectfully disagree that leveraging a frontier model constitutes a substantive weakness. Specifically, **(i)** the required frontier model usage imposes negligible overhead, and **(ii)** influential training pipelines in recent work explicitly rely on frontier-model guidance, filtering, or rewards.
>
> **(i) Minimal Cost Overhead.**
>
> As detailed in **Sec. 5 Minimal Compute Overhead**, the frontier model usage is a one-time cost that remains extremely small in practice (<$10 per benchmark). Moreover, because we are releasing both code and data at camera-ready time, the cost becomes effectively zero for anyone using our released benchmarks or replicating our experiments.
>
> **(ii) Leveraging Existing Foundation Models are Integral to Modern Foundation Model Training.**
>
> Relying on a frontier model is not only common but *foundational* to how state-of-the-art models are currently trained. Our approach follows this well-established paradigm rather than introducing any new dependency. Examples include:
>
> 1. RLAIF [1] leverage SoTA LLMs to provide reward to further tune the model.
> 2. Qwen and DeepSeek teams have explicitly stated that they curate their large-scale data via filtering using their previously SoTA model when training their next generation model [2, 3, 4, 5].
>
> ### **[W2] Clarification on How Tiny Proxy Models Work Well**
>
> In response to your comment:
> > it is unclear how the small proxy models less than <100M work so well
>
> Our paper provides a detailed explanation, and we summarize the key conceptual and empirical points here for clarity. First consider our objective in **Sec. 2 Problem Setting:**
>
> > Our objective is to design a proxy evaluation $\text{metric}^\text{p}$ such that considering $f: \text{metric}^\text{p} \mapsto \text{metric}^\text{t}$, find $\text{metric}^\text{p} := \max_{\text{metric}} \text{corr}(\text{metric}(\pi^\text{p}), \text{metric}^\text{t}(\pi^\text{t}))$
>
> Next, in **Sec. 3 Bridging Small and Large Model Scale Evaluation**, we granularly explain how small models can become strong proxies.
>
> We demonstrate why existing approaches fail at small model scale (**Sec. 3.1**):
> 1. Evaluation Objective Misalignment
>     2. Misaligned Evaluation Metric
>     3. Not All NLL are Equal: Distributional Alignment.
> 3. Target Task Misalignment
>
> Our claims are empirically validated across diverse settings (**Fig. 2**, **Fig. 3**, **Tab. 1**). These results show that once the misalignments are corrected, even <100M-parameter models produce highly correlated proxy evaluations despite their limited capacity.
>
> Motivated by these findings, **Sec. 3.2** introduces rBridge, a simple yet principled method for aligning evaluation objectives and tasks. We provide both intuitive visualization (**Fig. 1**) and concrete implementation details (**Appendix B**). We also show in **Tab. 1** and **Fig. 4** that our findings consistently improves proxy-to-target correlation and ranking across all evaluated dimensions.
>
> ### **[Q1] Added Compute Cost Measurement Method**
>
> In response to your comment:
> > How was the compute cost measured? Is it measured in a real system or just calculated by the model size?
>
> As noted in line 248 - 249, we follow the DataDecide protocol [6] and adopt the widely used FLOPs approximation FLOPs = $6ND$, originally introduced by OpenAI [7]. This formula is a commonly accepted proxy for compute cost, and applies consistently across all compared methods, allowing us to measure *relative* efficiency fairly.
>
> To avoid ambiguity, we now explicitly state in the main paper:
> > Compute cost is measured as FLOPs = $6ND$, where $N$ is model size, and $D$ is dataset size, following [7].
>
> We appreciate the suggestion, as it improves clarity for readers.

---

> ### Author Response · Authors · 2025-11-18
> **Response to Reviewer 4B33 (Part II)**
>
> ### **[Q2] Hardware and Software Details Available**
>
> In response to your comment:
> > What are the system (hardware and software) settings if a real system was implemented?
>
> We provide granular details on our hardware and software in **Appendix C Further Experimental Details**. We have included the hardware that we used in the **Appendix C.1 Hardware**. All training software follow commonly available ones as described in **Appendix C.2**, **C.3**, and **C.4**.
>
> As noted in line 450 and 553, we will also open-source both code and datasets by the camera-ready version, further supporting transparency and replicability.
>
> ### **[Q3] Proxy Performance Improves as we Scale Proxy Model Size**
>
> In response to your comment:
> > Using the proposed method, can a larger proxy model work better and how much better than a smaller proxy model?
>
> Yes, our results indicate that larger proxy models do generally yield better predictive performance. As shown in **Fig. 6 (a)**, the Pareto frontier of rBridge improves with increased proxy FLOPs, demonstrating that the proxy benefits from scale.
>
> However, a key observation is that naively scaling the proxy is insufficient. Even when we scale the proxy model by 7$\times$ and 13$\times$, when using the downstream metric (Acc./p@1; see **Fig. 7(a)**) it still falls short of our method. This highlights that our proposed approach provides substantial gains beyond what scaling alone can achieve, and remains advantageous even under proxy size enlargement.
>
> ### **Reference**
>
> [1] Lee et al. RLAIF vs. RLHF: Scaling Reinforcement Learning from Human Feedback with AI Feedback. ICML 2024
>
> [2] Qwen Team. Qwen2.5 Technical Report. 2025
>
> [3] Qwen Team. Qwen3 Technical Report. 2025
>
> [4] Qwen Team. Qwen2.5-VL Technical Report. 2025
>
> [5] DeepSeek-AI. DeepSeek-R1: Incentivizing Reasoning Capability in LLMs via Reinforcement Learning. 2025
>
> [6] Magnusson et al., AI2. DataDecide: How to Predict Best Pretraining Data with Small Experiments. ICML 2025
>
> [7] OpenAI. Scaling Laws for Neural Language Models. 2020

---

### Author Response · Authors · 2025-11-18
**Global Response**

We thank all reviewers and AC for their valuable time. We are especially encouraged to see that the reviewers recognize:
* Significant **Impact** and **Practical Value**
    * "... practically important ..." (`bXEE`)
    * "... great practical value ..." (`v8sd`)
    * "... very useful to the community for improving LLM pre-training efficiently ..." (`KWnN`)
    * "motivation and methodology are well thought through and sound" (`KWnN`)
* Significant Predictive **Performance** and **Compute Cost** Improvement
    * "significantly improved performance ..." (`4B33`)
    * "... reducing dataset ranking costs by over 100$\times$"(`bXEE`)
    * "... strong performance ..." (`bXEE`)
    * "... greatly reducing computational costs" (`bXEE`)
    * "... powerful new metric ..." (`KWnN`)
    * "results are very impressive ..." (`KWnN`)
* Innovative and Intellectually Engaging
    * "... intellectually engaging ..." (`bXEE`)
    * "... this work offers an innovative ..." (`KWnN`)
* Comprehensive Empirical Study
    * "empirical work is diligent and complete" (`KWnN`)
* Straight-forward Replication
    * "... easy to implement, with details well covered" (`v8sd`)

We responded to *each* review points to clear misunderstandings and improve our revised version. Note, additions or changes to the revised text have been highlighted in **$\textcolor{blue}{blue}$**, for reviewers, AC, and PC’s convenience.

We welcome further discussion for the advancement of the research community.

---

### Author Response · Authors · 2025-12-03
**Summary**

In light of recent events, we appreciate AC and PCs' significant additional effort for our community. We would like to summarize the rebuttal phase for their convenience. We believe that *all* concerns **[W]** and questions **[Q]** have been effectively addressed.

### **Strengths of our Contribution**
Please refer to our **Global Response**.

### **Factual Errors in Review**

We reiterate that we are particularly concerned about **factual errors** in some of the reviews. For instance,
- We stayed within the 9 page limit at submission
- Our work already includes a coding benchmark

which reviewer `bXEE` states otherwise.

### **Response and Revision:** `KWnN`

In summary `KWnN` states "I am happy with the revisions".

| Overview | Response Type |
| -------- | -------- |
| [W1] Added Additional References     |Writing Updated |
| [W2] Fixed Minor Spelling and Syntax Suggestions |Writing Updated |
| [W3] Added Limitation and Future Directions to Main Body |Writing Updated |
| [W4] Commited to Open Source and Open Data |Writing Updated |
| [W5] We Agree that our Method Helps The Environment |Writing Updated |
| [Q1] Letter-level Probabilities are Conditioned on Previous Tokens | Answered |
| [Q2] $R^\phi$ $\rightarrow$ rBridge Performance is Not Negligable |Answered |

### **Response and Revision:** `v8sd`

`v8sd` responded with "Thanks for your clarification".

| Overview | Response Type |
| -------- | -------- |
| [W1] Pre-training Benchmarking on Tasks that Frontier Models Fails At is Lower Priority    | Addressed with 5 References |
| [W2] Ablation Study of Weighted NLL Already Exists    | Already Exists |

### **Response and Revision:** `bXEE`

`bXEE` did not respond.

| Overview | Response Type |
| -------- | -------- |
| [W1] We Have Analyzed Why Some Benchmark Datasets are Noisy at Small Model Scale  | Addressed with Reference to our Paper |
| [W2] Our Method is Frontier Model Choice Agnostic  | Addressed with Reference to our Paper and the Literature |
| [W3] Coding Reasoning Benchmark is Not Missing; and Subjective Benchmarks are Out-of-scope  | Already Exists; Addressed with 3 References |
| [W4] Added Additional Curve Fitting Example Visualizations | Writing Updated; +10 Figs. Added |
| [W5] Fixed Minor Typos | Writing Updated |
| [W6] Paper Does Not Exceed 9 Pages (Ethics/Reproducibility Statement Does Not Count)| Incorrect Review |

### **Response and Revision:** `4B33`

`4B33` did not respond.

| Overview | Response Type |
| -------- | -------- |
| [W1] Leveraging an Existing Frontier-model to Advance Next Generation Foundation Model Training is Common Practice  | Addressed with 5 References |
| [W2] Clarification on How Tiny Proxy Models Work Well | Addressed with Reference to our Work |
| [Q1] Added Compute Cost Measurement Method | Answered; Writing Updated |
| [Q2] Hardware and Software Details Available | Answered |
| [Q3] Proxy Performance Improves as we Scale Proxy Model Size | Answered |

---

### Meta-Review · Area_Chair_NeDD · 2026-01-19

**Summary:**

This paper proposes rBridge, a proxy-evaluation method that uses small models to predict large-model reasoning performance by scoring frontier-model-generated reasoning traces with a task-aligned weighted NLL. Reviewers agree the problem is practically important and find the approach generally effective with extensive experiments.

The main concerns before the rebuttal are:
(1) dependence on a frontier model to generate the “gold” reasoning traces, (2) clarity/justification for why very small proxies (<100M) work well, (3) evaluation breadth and robustness, (4) missing ablations/clarity (e.g., contribution of weighted NLL; compute-cost measurement details), and (5) missing predictability/assessor-model references

**Reviewer Concerns:**

Reviewer 4B33: questioned compute-cost measurement, setup details, proxy scaling behavior, and why very small proxies work; authors clarified cost accounting and provided more detail/argumentation about proxy predictiveness and scaling.

Reviewer bXEE: asked for stronger analysis of noisy benchmarks/dataset characteristics, sensitivity to frontier-model choice, broader visualizations beyond a single example, and breadth beyond math/QA, plus minor presentation issues; authors added/expanded discussion, clarified code benchmark coverage, included more plots, and fixed presentation points.

Reviewer v8sd: raised concerns about frontier-trace quality and requested clearer ablation for weighted NLL; authors clarified scope and pointed to (or added) ablation evidence; reviewer acknowledged the clarification.

Reviewer KWnN: focused on missing related work (predictability/assessor-model style literature), writing clarity, and bringing limitations into main text; authors updated accordingly.

Still concerns: the approach still fundamentally depends on having a strong model to generate traces; skepticism may remain for settings where trace quality is poor or frontier-model choice matters.

**Reviewer Scores:**

Reviewer bXEE may increase scores: Many concrete concerns appear addressed (analysis/plots/coverage/presentation), but residual uncertainty around frontier dependence and broader generalization could keep it borderline.

Reviewer 4B33 may increase scores. Clarifications help cost measurement and proxy rationale

Overall, I think the paper may be possible accepted as borderline positive.

---

### Decision · Program_Chairs · 2026-01-26

Accept (Poster)